# Utility-Diversity Aware Online Batch Selection for LLM Supervised Fine-tuning

## Abstract

Supervised fine-tuning (SFT) is a commonly used technique to adapt large language models (LLMs) to downstream tasks. In practice, SFT on a full dataset is computationally expensive and sometimes suffers from overfitting or bias amplification. This facilitates the rise of data curation in SFT, which prioritizes the most valuable data to optimze. This work studies the online batch selection family that dynamically scores and filters samples during the training process. However, existing popular methods often (i) rely merely on the utility of data to select a subset while neglecting other crucial factors like diversity, (ii) rely on external resources such as reference models or validation sets, and (iii) incur extra training time over full-dataset training. To address these limitations, this work develops **UDS (Utility-Diversity Sampling)**, a framework for efficient online batch selection in SFT. UDS leverages the nuclear norm of the logits matrix to capture both data utility and intra-sample diversity, while estimating inter-sample diversity through efficient low-dimensional embedding comparisons with a lightweight memory buffer of historical samples. Such a design eliminates the need for external resources and unnecessary backpropagation, securing computational efficiency. Experiments on multiple benchmarks demonstrate that UDS consistently outperforms state-of-the-art online batch selection methods under varying data budgets, and significantly reduces training time compared to full-dataset fine-tuning.

## 1 Introduction

The rapid progress of Large Language Models (LLMs) has reshaped natural language processing and enabled impressive generalization across a wide range of domains (Achiam et al., 2023; Brown et al., 2020; Liu et al., 2024; Touvron et al., 2023; Bai et al., 2023). To further improve their performance in specialized areas, Supervised Fine-tuning (SFT) has emerged as a typical post-training paradigm. However, training on a full dataset is not always advantageous. It entails substantial computational cost and has been shown to be less effective than using a small amount of carefully curated data (Albalak et al., 2024; Xia et al., 2024; Zhou et al., 2023). These challenges highlight the importance of principled data selection strategies that can improve both efficiency and effectiveness in SFT.

In this work, we address this challenge by focusing on **online batch selection**, a paradigm that dynamically evaluates sample value and performs filtering during training (Wang et al., 2024; Loshchilov & Hutter, 2015; Katharopoulos & Fleuret, 2018; Mindermann et al., 2022). Instead of using all samples in a batch, the model assesses the importance of each sample as it is encountered and selects only a subset to participate in parameter updates. This approach enables the selection process to adapt in real time to the model's current state and learning trajectory, improving both training efficiency and effectiveness.

However, despite promising progress, current online batch selection methods still face several issues that hinder their practical deployment. These methods typically rely solely on *data utility*, e.g., picking sample subsets with high loss (Loshchilov & Hutter, 2015; Jiang et al., 2019) or gradient magnitude (Katharopoulos & Fleuret, 2018). While such heuristics enable capturing the critical samples to reduce the current training loss, they evaluate sample value from a limited perspective. In practice, effective selection also requires considerations of *intra-sample diversity*, to reflect the richness of information with fewer repetitive phrases within each training instance, and *inter-sample diversity* to suppress redundancy across examples by avoiding repeated training on near-duplicate content (Lee et al., 2021; Tirumala et al., 2023). Popular literature (Loshchilov & Hutter, 2015;

Table 1: **Comparison of online batch selection methods under our desiderata (D1–D3).** The columns denote: Data Utility, Intra-sample Div. (Intra-sample Diversity), Inter-sample Div. (Inter-sample Diversity), No External Res. (No External Resources), and Train. Time Reduc. (Training Time Reduction). The symbols denote: ✓ (support), and ✗ (don't support).

| Method | Data Utility | Intra-sample Div. | Inter-sample Div. | No External Res. | Train. Time Reduc. |
|---|---|---|---|---|---|
| Max Loss | ✓ | ✗ | ✗ | ✓ | ✓ |
| Max Grad | ✓ | ✗ | ✗ | ✓ | ✗ |
| RHO-Loss | ✓ | ✗ | ✗ | ✗ | ✗ |
| GREATS | ✓ | ✗ | ✓ | ✗ | ✗ |
| **UDS (Ours)** | ✓ | ✓ | ✓ | ✓ | ✓ |

Katharopoulos & Fleuret, 2018; Mindermann et al., 2022) prioritizes data utility and rarely examines the selection criteria through the lens of diversity.

Another primary issue involves both *external dependencies* and *computational overhead*. Some approaches (Mindermann et al., 2022; Wang et al., 2024) rely on a held-out validation set, which may be unavailable since we hardly know the exact distribution of test dataset. Other methods (Mindermann et al., 2022; Deng et al., 2023) use a reference model that is also shown to be impractical in the real world (Kaddour et al., 2023). Besides these external dependencies, several selection algorithms (Katharopoulos & Fleuret, 2018; Wang et al., 2024; Mindermann et al., 2022) introduce substantial computational costs that even exceed the expense of full-dataset training, raising efficiency concerns.

Based on the above evidence, this work proposes that an ideal online batch selection method should be comprehensive and satisfy the three desiderata (D1-D3) outlined below. A systematic comparison of representative methods against these criteria is summarized in Table 1.

**D1:** Jointly consider *data utility*, *intra-sample diversity*, and *inter-sample diversity*;

**D2:** Circumvent the access to external resources, e.g., reference model or validation set;

**D3:** Reduce the training time of the overall pipeline relative to the full-dataset SFT.

**Leveraging logits as basis for online scoring.** To operationalize these desiderata, a key challenge lies in how to efficiently and reliably score candidate samples during training. In online batch selection, each candidate sample must be evaluated under the current model to quantify its contribution to learning. This evaluation typically requires at least a forward pass, as it can fully capture how the model presently interprets the sample and thus provide an accurate basis for selection (Huang et al., 2023; Shorinwa et al., 2025). To achieve this efficiently (D3) without external resources (D2), we must avoid expensive gradient computations for each candidate sample and leverage intrinsic signals already available during forward propagation. These considerations collectively justify leveraging the model's output logits, which are naturally produced during the forward pass and encode rich information about both sample utility and diversity (Qiu & Miikkulainen, 2024; Geng et al., 2023). Specifically, we compute an *intra-sample importance score* based on the nuclear norm of the logits, which captures the utility and intra-sample diversity of a sample, and an *inter-sample importance score* using low-dimensional similarity matching against previously selected samples to quantify inter-sample diversity (D1). Integrating these scores allows the training process to balance exploitation of high-utility samples with exploration of less-visited regions in the data distribution, thereby improving the overall effectiveness of online batch selection.

In summary, we present **UDS (Utility-Diversity Sampling)**, a new framework for online batch selection in SFT of LLMs. Our main contributions are:

1. To capture intra-sample value, we leverage the nuclear norm of the logits matrix, which naturally reflects both *data utility* and *intra-sample diversity*, providing a principled criterion for assessing within-sample informativeness.

2. To estimate inter-sample diversity, we design a structured bilinear random projection of logits for compact embedding and show that similarity matching with historical data points effectively reduces redundancy with negligible overhead.

3. Our UDS operates directly on forward-pass outputs without external resources, achieving faster convergence than full-dataset SFT and surpassing existing online batch selection baselines.

## 2 PRELIMINARIES

### 2.1 PROBLEM FORMULATION OF ONLINE BATCH SELECTION

At training iteration $t$, we draw a candidate batch $\mathcal{B}_t = \{(\boldsymbol{x}_t^i, y_t^i)\}_{i=1}^B$ from the corpus. The goal of online batch selection is to choose a subset $\widehat{\mathcal{B}}_t \subseteq \mathcal{B}_t$ that maximizes the immediate (or near-term) benefit to the model. Let $s(\boldsymbol{x}_t^i; \boldsymbol{\theta}_t, \mathcal{H}_t) \in \mathbb{R}$ denote a per-sample importance score computed under the current model state $\boldsymbol{\theta}_t$ and optional history $\mathcal{H}_t$ (e.g., a small buffer of recent embeddings). The selection objective can be viewed as an optimization problem:

$$\widehat{\mathcal{B}}_t = \arg \max_{S \subseteq \mathcal{B}_t, |S| = K} \sum_{i \in S} s(\boldsymbol{x}_t^i; \boldsymbol{\theta}_t, \mathcal{H}_t), \tag{1}$$

where $K$ controls the fraction of data we aim to select.

### 2.2 AUTOREGRESSIVE LANGUAGE MODELING

Our work focuses on autoregressive language models that factorize the sequence probability via the chain rule:

$$p_{\boldsymbol{\theta}_t}(\boldsymbol{x}_t^i) = \prod_{n=1}^N p_{\boldsymbol{\theta}_t}(x_t^{i,n} \mid \boldsymbol{x}_t^{i,<n}), \tag{2}$$

where $\boldsymbol{x}_t^{i,<n} = (x_t^{i,1}, \ldots, x_t^{i,n-1})$ denotes the context prefix, and $\boldsymbol{\theta}_t \in \mathbb{R}^p$ represents model parameters at iteration $t$. At each generation step $n$, model $\pi_{\boldsymbol{\theta}_t}$ produces a logit vector $\boldsymbol{l}_t^{i,n} \in \mathbb{R}^V$ over vocabulary $\mathcal{V}$, with $V = |\mathcal{V}|$ the vocabulary size. For a sequence of length $N$, these vectors $\boldsymbol{l}_t^{i,1}, \ldots, \boldsymbol{l}_t^{i,N}$ collectively form the logits matrix $\boldsymbol{L}(\boldsymbol{x}_t^i; \boldsymbol{\theta}_t) \in \mathbb{R}^{N \times V}$, where the $n$-th row corresponds to logits at position $n$. $\boldsymbol{L}(\boldsymbol{x}_t^i; \boldsymbol{\theta}_t)$ captures the model's predictive distribution at each position. Applying the softmax function to each row of $\boldsymbol{L}(\boldsymbol{x}_t^i; \boldsymbol{\theta}_t)$ yields the conditional probability distribution for the next token. The resulting probability matrix is denoted $\boldsymbol{P}(\boldsymbol{x}_t^i; \boldsymbol{\theta}_t) \in \mathbb{R}^{N \times V}$, with entries

$$p_{\boldsymbol{\theta}_t}(x_t^{i,n} \mid \boldsymbol{x}_t^{i<n}) = \frac{\exp\big(\pi_{\boldsymbol{\theta}_t}(\boldsymbol{x}_{<n})[x_n]\big)}{\sum_{y \in \mathcal{V}} \exp\big(\pi_{\boldsymbol{\theta}_t}(\boldsymbol{x}_{<n})[y]\big)}. \tag{3}$$

During fine-tuning, the parameters $\boldsymbol{\theta}_t$ are updated to maximize the likelihood of training sequences. At iteration $t$, the online selection mechanism identifies an informative subset $\widehat{\mathcal{B}}_t$ from the candidate batch $\mathcal{B}_t$, which is then used to perform a gradient update $\boldsymbol{\theta}_t \to \boldsymbol{\theta}_{t+1}$.

## 3 UTILITY-DIVERSITY SAMPLING (UDS)

We propose **UDS**, an online batch selection framework that jointly considers data utility, intra-sample diversity, and inter-sample diversity. It scores and samples each data point using a mixture of two complementary scores: (i) the *Nuclear Norm* of the logits matrix ($s_{\text{intra}}^{t,i}$), capturing optimization utility and intra-sample diversity; and (ii) the *Diversity Distance* ($s_{\text{inter}}^{t,i}$), measuring dispersion against recent selections. An overview is shown in Figure 1, with pseudocode provided in Algorithm 1.

### 3.1 INTRA-SAMPLE IMPORTANCE SCORE VIA NUCLEAR NORM

The intra-sample importance score ($s_{\text{intra}}^{t,i}$) can be characterized through two complementary views: (i) *optimization utility*—how much the sample can contribute to loss reduction during training, and (ii) *intra-sample diversity*—how diverse the model's token-level outputs are within the sequence. We employ the nuclear norm (trace norm) of the logits matrix $\boldsymbol{L}(\boldsymbol{x}_t^i; \boldsymbol{\theta}_t)$ to capture both aspects. The nuclear norm $\|\boldsymbol{L}(\boldsymbol{x}_t^i; \boldsymbol{\theta}_t)\|_*$ is defined as the sum of singular values of $\boldsymbol{L}(\boldsymbol{x}_t^i; \boldsymbol{\theta}_t)$. Let $\{\sigma_1, \sigma_2, \ldots, \sigma_r\}$ denote its singular values (with $r = \text{rank}\big(\boldsymbol{L}(\boldsymbol{x}_t^i; \boldsymbol{\theta}_t)\big)$), then $s_{\text{intra}}^{t,i}$ is computed as:

$$s_{\text{intra}}^{t,i} = \|\boldsymbol{L}(\boldsymbol{x}_t^i; \boldsymbol{\theta}_t)\|_* = \sum_{j=1}^r \sigma_j. \tag{4}$$

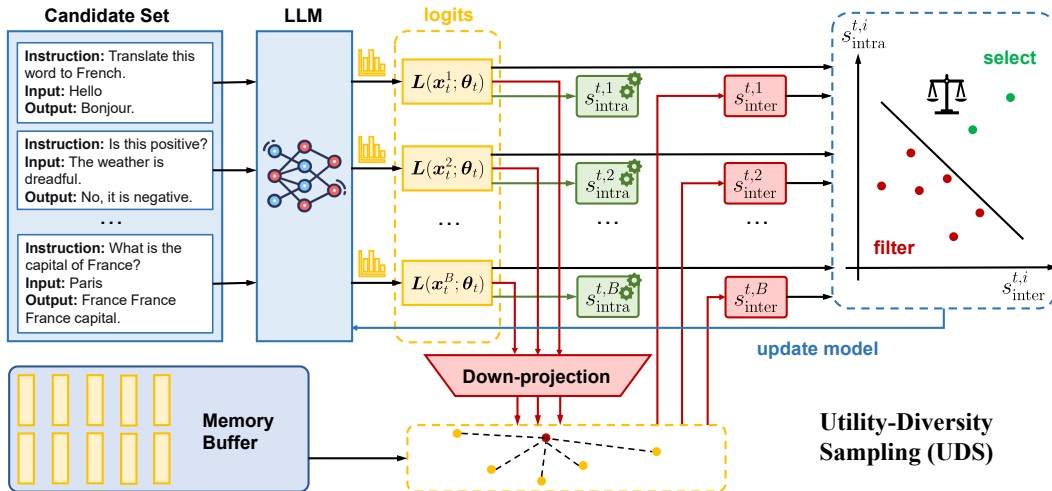

Figure 1: **Schematic of the UDS Framework.** In the forward pass, the LLM extract logits $L(x_t^i; \theta_t)$ for each sample. Using $L(x_t^i; \theta_t)$, we first calculate its nuclear norm for the intra-sample importance score $s_{\text{intra}}^{t,i}$, and then down-projected to calculate the distance $s_{\text{inter}}^{t,i}$ with historical samples in the memory buffer. Finally, we select the top-$K$ valuable samples to update the LLM.

In mathematics, the nuclear norm and the Frobenius norm can bound each other, as shown in Lemma 3.1. A detailed explanation is provided in Appendix C.1.

**Lemma 3.1.** *(Horn & Johnson, 2012) For any matrix $L(x_t^i; \theta_t) \in \mathbb{R}^{N \times V}$, the following inequality holds:*

$$\|L(x_t^i; \theta_t)\|_F \leq \|L(x_t^i; \theta_t)\|_* \leq \sqrt{\min(N, V)} \|L(x_t^i; \theta_t)\|_F,$$

*where $\| \cdot \|_F$ denotes the Frobenius norm, computed as the square root of the sum of squares of all entries in the matrix. The left inequality achieves equality when $L(x_t^i; \theta_t)$ is rank-1 and has only one non-zero singular value. The right inequality achieves equality when $L(x_t^i; \theta_t)$ has full rank and all singular values are equal.*

Thus, a larger nuclear norm arises in two ways: (1) the Frobenius norm increases, indicating larger logits, and (2) for a fixed Frobenius norm, the nuclear norm shifts closer to the upper bound. Below we clarify how these two effects embody the two complementary aspects of sample-importance score.

**Larger Nuclear Norm Positively Correlates With Higher Optimization Utility.** Traditional notions of sample difficulty—such as *maximum loss* (Loshchilov & Hutter, 2015) or *maximum gradient* (Katharopoulos & Fleuret, 2018)—often misalign with actual training dynamics (Wang et al., 2024). Instead, we characterize sample importance through its potential contribution to loss reduction, termed its *optimization utility*. For ease of exposition, we assume training with SGD using batch size $B$ and a learning rate $\eta_t$. An extension to Adam is provided in Appendix C.2. The parameter update at iteration $t$ is $\theta_{t+1} - \theta_t = -\frac{\eta_t}{B} \sum_{j=1}^{B} \nabla_\theta \ell(x_t^j; \theta_t)$. After the update, the logits matrix for datapoint $x_t^i$ can be expanded using a first-order Taylor expansion:

$$L(x_t^i; \theta_{t+1}) \approx L(x_t^i; \theta_t) + \nabla_\theta L(x_t^i; \theta_t) \cdot (\theta_{t+1} - \theta_t), \tag{5}$$

so that the induced change is

$$\delta L(x_t^i; \theta_t) = L(x_t^i; \theta_{t+1}) - L(x_t^i; \theta_t) \approx -\frac{\eta_t}{B} \nabla_\theta L(x_t^i; \theta_t) \cdot \left( \sum_{j=1}^{B} \nabla_\theta \ell(x_t^j; \theta_t) \right). \tag{6}$$

---

**Algorithm 1** Utility-Diversity Sampling (UDS)

---

**Input:** Candidate batch $\mathcal{B}_t$ ($t = 0, 1, \ldots, T$), initial model $\pi_{\boldsymbol{\theta}_0}$, memory queue $\boldsymbol{Q}$

1: Initialize memory queue $\boldsymbol{Q} \leftarrow \{\}$
2: **for** $t = 0, 1, \ldots, T$ **do**
3:     // Calculate importance score and dynamically sampling
4:     **for** each sample $\boldsymbol{x}_t^i \in \mathcal{B}_t$ **do**
5:         Perform forward pass to obtain logits $\boldsymbol{L}(\boldsymbol{x}_t^i; \boldsymbol{\theta}_t) \leftarrow \pi_{\boldsymbol{\theta}_t}(\boldsymbol{x}_t^i)$
6:         Calculate intra-sample importance score $s_{\text{intra}}^{t,i} \leftarrow \|\boldsymbol{L}(\boldsymbol{x}_t^i; \boldsymbol{\theta}_t)\|_*$
7:         Randomly project to low-dimensional embeddings $\boldsymbol{z}_t^i \leftarrow \text{vec}(\boldsymbol{\Gamma}_2 \cdot \boldsymbol{L}(\boldsymbol{x}_t^i; \boldsymbol{\theta}_t) \cdot \boldsymbol{\Gamma}_1^\top)$
8:         Calculate inter-sample importance score $s_{\text{inter}}^{t,i} \leftarrow \frac{1}{|\boldsymbol{Q}|} \sum_{\boldsymbol{z}_j \in \boldsymbol{Q}} \|\boldsymbol{z}_t^i - \boldsymbol{z}_j\|_2$
9:         Combine intra- and inter-sample importance scores $s_{\text{total}}^{t,i} = s_{\text{intra}}^{t,i} + \alpha * s_{\text{inter}}^{t,i}$
10:     **end for**
11:     Dynamically select top-$K$ samples $\widehat{\mathcal{B}}_t \leftarrow \text{TopK}(\{s_{\text{total}}^{t,i}\}_{i=1}^B, K)$
12:     // Update memory queue
13:     **while** $|\boldsymbol{Q}| + K > M$ **do**
14:         $\boldsymbol{Q} \leftarrow \boldsymbol{Q} \setminus \{\text{oldest element}\}$
15:     **end while**
16:     $\boldsymbol{Q} \leftarrow \boldsymbol{Q} \cup \{\boldsymbol{z}_t^i \mid \boldsymbol{x}_t^i \in \widehat{\mathcal{B}}_t\}$
17:     Update model parameters $\boldsymbol{\theta}_{t+1} \leftarrow \boldsymbol{\theta}_t$ through backpropagation
18: **end for**

---

Rather than reasoning in parameter space, we can view the training step as producing a logits perturbation $\delta \boldsymbol{L}(\boldsymbol{x}_t^i; \boldsymbol{\theta}_t)$. The first-order Taylor expansion of the loss around $\boldsymbol{L}(\boldsymbol{x}_t^i; \boldsymbol{\theta}_t)$ gives

$$\delta\ell(\boldsymbol{x}_t^i; \boldsymbol{\theta}_t) := \ell\left(\boldsymbol{L}(\boldsymbol{x}_t^i; \boldsymbol{\theta}_t) + \delta\boldsymbol{L}(\boldsymbol{x}_t^i; \boldsymbol{\theta}_t)\right) - \ell\left(\boldsymbol{L}(\boldsymbol{x}_t^i; \boldsymbol{\theta}_t)\right) \approx \langle \nabla_{\boldsymbol{L}}\ell(\boldsymbol{L}(\boldsymbol{x}_t^i; \boldsymbol{\theta}_t)), \delta\boldsymbol{L}(\boldsymbol{x}_t^i; \boldsymbol{\theta}_t)\rangle, \quad (7)$$

where $\langle \cdot, \cdot \rangle$ denotes the Frobenius inner product, and $\nabla_{\boldsymbol{L}}\ell(\boldsymbol{L}(\boldsymbol{x}_t^i; \boldsymbol{\theta}_t))$ is the gradient matrix of $\ell$ with respect to $\boldsymbol{L}$. For cross-entropy loss, $\nabla_{\boldsymbol{L}}\ell(\boldsymbol{L}(\boldsymbol{x}_t^i; \boldsymbol{\theta}_t)) = \boldsymbol{P}(\boldsymbol{x}_t^i; \boldsymbol{\theta}_t) - \boldsymbol{Y}(\boldsymbol{x}_t^i)$, which we denote as $\boldsymbol{\Delta}(\boldsymbol{x}_t^i; \boldsymbol{\theta}_t)$. Hence,

$$\delta\ell(\boldsymbol{x}_t^i; \boldsymbol{\theta}_t) \approx \langle \boldsymbol{\Delta}(\boldsymbol{x}_t^i; \boldsymbol{\theta}_t), \delta\boldsymbol{L}(\boldsymbol{x}_t^i; \boldsymbol{\theta}_t)\rangle. \quad (8)$$

Intuitively, as the Frobenius norm of the logits $\|\boldsymbol{L}(\boldsymbol{x}_t^i; \boldsymbol{\theta}_t)\|_F$ increases, the perturbation $\delta\boldsymbol{L}(\boldsymbol{x}_t^i; \boldsymbol{\theta}_t)$

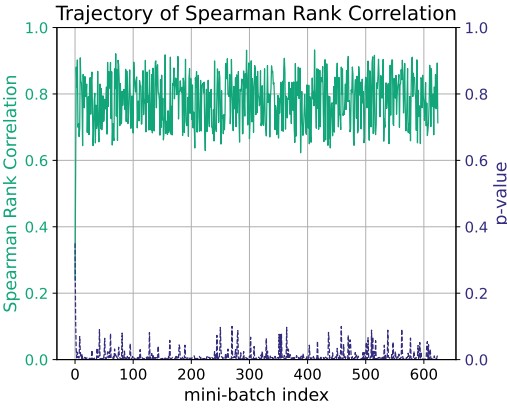
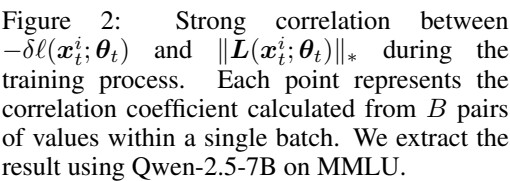
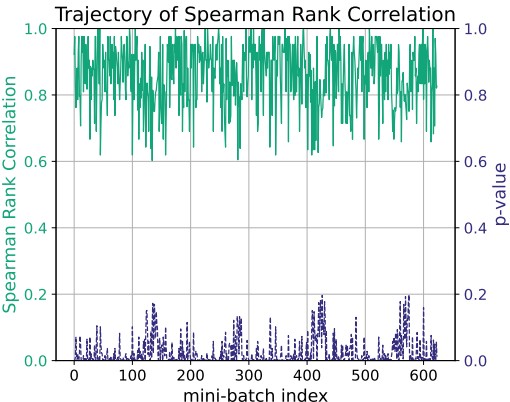

Figure 2: Strong correlation between $-\delta\ell(\boldsymbol{x}_t^i; \boldsymbol{\theta}_t)$ and $\|\boldsymbol{L}(\boldsymbol{x}_t^i; \boldsymbol{\theta}_t)\|_*$ during the training process. Each point represents the correlation coefficient calculated from $B$ pairs of values within a single batch. We extract the result using Qwen-2.5-7B on MMLU.

Figure 3: Strong correlation between $rank(\boldsymbol{L}(\boldsymbol{x}_t^i; \boldsymbol{\theta}_t))$ and $\|\boldsymbol{L}(\boldsymbol{x}_t^i; \boldsymbol{\theta}_t)\|_*$ during the training process. Each point represents the correlation coefficient calculated from $B$ pairs of values within a single batch. We extract the result using Qwen-2.5-7B on MMLU.

also tends to grow uniformly. This occurs because each entry in the activation logits becomes larger, and these values are directly used during backpropagation. In contrast, $\boldsymbol{\Delta}(\boldsymbol{x}_t^i; \boldsymbol{\theta}_t)$ is less sensitive to scale because it depends only on the normalized probabilities and the one-hot labels. Therefore,

$\|\boldsymbol{L}(\boldsymbol{x}_t^i; \boldsymbol{\theta}_t)\|_F$, $\|\boldsymbol{L}(\boldsymbol{x}_t^i; \boldsymbol{\theta}_t)\|_*$, and the attainable loss reduction $-\delta\ell(\boldsymbol{x}_t^i; \boldsymbol{\theta}_t)$ typically grow in tandem. Empirically, Figure 2 confirms a strong correlation between a sample's loss reduction and its nuclear norm. Hence, the nuclear norm can serve as an indicator of a sample's optimization utility.

When a single sample induces a larger loss reduction, it indicates that the sample both challenges the model's current predictions and provides informative gradients that effectively guide parameter updates. Such samples are particularly valuable for data selection: they not only highlight under-learned regions of the input space, but also accelerate training by yielding stronger optimization signals compared to redundant or already well-learned samples. We provide a categorization of the loss before and after training in Table 2. Hence, we prioritize selecting samples with larger nuclear norm that are more likely to have higher loss reduction.

Table 2: **Categorization of training samples by initial and final loss.** The optimization utility depends on how much the training loss decreases after training. Green entries denote high-utility samples that should be preferentially selected, while Red entries denote low-utility ones that should generally be avoided.

| Before/After | High | Low |
|---|---|---|
| **High** | Too Hard | Informative |
| **Low** | Overfitted | Too Easy |

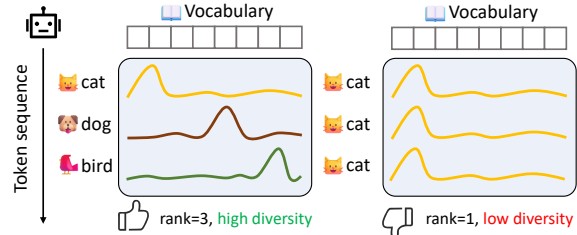

Figure 4: **Illustration of Intra-sample Diversity. Left:** Token sequence with high diversity, where the model predicts varied tokens (cat, dog, bird). **Right:** Token sequence with low diversity, where the model predominantly predicts a single token (cat).

**Larger Nuclear Norm Positively Correlates With Higher Intra-sample Diversity.** Beyond optimization utility, a larger nuclear norm can also arise from the structural diversity of token-level output distributions within a sequence. Recall Lemma 3.1: for a fixed Frobenius norm, the nuclear norm achieves its *minimum* when $\boldsymbol{L}(\boldsymbol{x}_t^i; \boldsymbol{\theta}_t)$ is rank-1 with only one singular value, and its *maximum* when $\boldsymbol{L}(\boldsymbol{x}_t^i; \boldsymbol{\theta}_t)$ has full rank with equal singular values. These two extremes correspond to distinct regimes of intra-sample diversity. The lower bound typically occurs when the model predicts only a single vocabulary token throughout the sequence (indicating repetitive generation behavior), while the upper bound is achieved when the model produces a florid and diverse vocabulary distribution across tokens (reflecting richer semantic information content). Figure 4 presents a simplified example illustrating what intra-sample diversity means in real cases. Intuitively, A larger nuclear norm implies that more vocabularies in this sequence are predicted and utilized, indicating higher prediction dispersity. We additionally provide an empirically strong correlation between $\|\boldsymbol{L}(\boldsymbol{x}_t^i; \boldsymbol{\theta}_t)\|_*$ and $rank(\boldsymbol{L}(\boldsymbol{x}_t^i; \boldsymbol{\theta}_t))$ in Figure 3 to support this. Specific explanations are listed below.

**Low Nuclear Norm Positively Correlates with Low rank and collinear rows (low diversity).** Considering the lower bound of Lemma 3.1, since $\boldsymbol{L}(\boldsymbol{x}_t^i; \boldsymbol{\theta}_t)$ is rank-1, the row vectors $\boldsymbol{l}_n$ are nearly collinear, i.e., $\boldsymbol{l}_n \approx \alpha_n \boldsymbol{v}$ for some fixed $\boldsymbol{v}$. This degenerate spectral structure indicates minimal diversity: the model's predictions for different tokens are aligned along the same direction, resulting in repetitive output.

**High Nuclear Norm Positively Correlates with High rank and orthogonal rows (high diversity).** Considering the higher bound of Lemma 3.1, $\boldsymbol{L}(\boldsymbol{x}_t^i; \boldsymbol{\theta}_t)$ has full rank and all singular values are approximately equal. This occurs when the row vectors are diverse and orthogonal with comparable norms. This flat spectrum indicates maximal diversity: each token is semantically meaningful to the output, and predictions are spread across diverse directions, capturing a wide range of semantic information.

## 3.2 Inter-Sample Importance Score via Low-rank Similarity Matching

Current online batch selection methods (Wang et al., 2024) only consider sample diversity within the candidate batch, which is suboptimal because the capacity of the batch is much smaller than the global datasets. To enhance global diversity, we maintain a fixed-size First-In-First-Out (FIFO) memory buffer $\boldsymbol{Q} \in \mathbb{R}^{M \times d}$, which stores representations $\boldsymbol{z}_t^i \in \mathbb{R}^d$ of the last $M$ samples selected

for training ($M \gg B$), and measure inter-sample diversity by calculating the distance between the candidate sample and the recent training history.

**Diversity Distance:** The inter-sample diversity score for the $i$-th sample is computed as its average Euclidean distance to all representations in the memory buffer:

$$s_{\text{inter}}^{t,i} = \frac{1}{|\boldsymbol{Q}|} \sum_{\boldsymbol{z}_j \in \boldsymbol{Q}} \|\boldsymbol{z}_t^i - \boldsymbol{z}_j\|_2. \tag{9}$$

If $\boldsymbol{Q}$ is empty, $s_{\text{inter}}^{t,i}$ is set to zero. A high $s_{\text{inter}}^{t,i}$ indicates a large distinction from recent data.

**Low-dimensional Projection.** As mentioned, the logits matrix $\boldsymbol{L}(\boldsymbol{x}_t^i; \boldsymbol{\theta}_t) \in \mathbb{R}^{N \times V}$ is semantically meaningful for computing diversity, but directly storing the whole matrix in the memory buffer is prohibitive (e.g., $\sim 74$GB for 1024 samples in Qwen-2.5-7B). A natural strategy is to randomly project it into low-dimensional vectors $\boldsymbol{z}_t^i \in \mathbb{R}^d$, while preserving pairwise distances (Johnson et al., 1984). However, a direct projection using $\boldsymbol{\Gamma} \in \mathbb{R}^{NV \times d}$ also incurs severe storage cost for the down-projection matrix ($\sim 74$GB in Qwen-2.5-7B if $d$ is set to 1024). To avoid this, we try to factorize $\boldsymbol{\Gamma}$ into two smaller projections: $\boldsymbol{\Gamma}_1 \in \mathbb{R}^{d_1 \times V}$ reducing the vocabulary dimension and $\boldsymbol{\Gamma}_2 \in \mathbb{R}^{d_2 \times N}$ reducing the sequence length dimension, which together approximate the original projection with $d = d_1 d_2$. Formally, we aim to find proper $\boldsymbol{\Gamma}_1$ and $\boldsymbol{\Gamma}_2$ that operate in the following way:

$$\boldsymbol{z}_t^i = \text{vec}\left(\boldsymbol{\Gamma}_2 \cdot \boldsymbol{L}(\boldsymbol{x}_t^i; \boldsymbol{\theta}_t) \cdot \boldsymbol{\Gamma}_1^\top\right), \tag{10}$$

Fortunately, the *subsampled randomized Fourier transform* (SRFT)-style construction for $\boldsymbol{\Gamma}_1$ and $\boldsymbol{\Gamma}_2$ (see Theorem 3.2) can fulfill this role (Ailon & Chazelle, 2006; Jin et al., 2021). This approach achieves an approximate Johnson-Lindenstrauss embedding while avoiding the storage of an explicit $NV \times d$ projection matrix and reduces the computational complexity from $\mathcal{O}(NVd)$ to $\mathcal{O}\left((N + V)d \log(NV)\right)$. The complete proof is provided in Appendix C.3.

---

**Theorem 3.2.** *Let $N, V \in \mathbb{N}$, and choose $d_1 \leq V$, $d_2 \leq N$ with $d = d_1 d_2$. Define*

$$\boldsymbol{\Gamma}_1 = \sqrt{\frac{V}{d_1}} \boldsymbol{S}_1 \boldsymbol{F}_1 \boldsymbol{D}_1 \in \mathbb{R}^{d_1 \times V}, \qquad \boldsymbol{\Gamma}_2 = \sqrt{\frac{N}{d_2}} \boldsymbol{S}_2 \boldsymbol{F}_2 \boldsymbol{D}_2 \in \mathbb{R}^{d_2 \times N},$$

*where $\boldsymbol{F}_1, \boldsymbol{F}_2$ are orthonormal transforms (discrete Fourier transform (DFT) matrices), $\boldsymbol{D}_1, \boldsymbol{D}_2$ are random $\{\pm 1\}$ diagonal matrices with independent Rademacher entries on the diagonal, and $\boldsymbol{S}_1, \boldsymbol{S}_2$ are random selection matrices that choose $d_1$ and $d_2$ rows uniformly at random without replacement. For each $\boldsymbol{L}(\boldsymbol{x}_t^i; \boldsymbol{\theta}_t) \in \mathbb{R}^{N \times V}$, define*

$$\boldsymbol{u}_t^i = \text{vec}(\boldsymbol{L}(\boldsymbol{x}_t^i; \boldsymbol{\theta}_t)) \in \mathbb{R}^{NV}, \qquad \boldsymbol{v}_t^i = \text{vec}(\boldsymbol{\Gamma}_2 \cdot \boldsymbol{L}(\boldsymbol{x}_t^i; \boldsymbol{\theta}_t) \cdot \boldsymbol{\Gamma}_1^\top) \in \mathbb{R}^d.$$

*Then there exists a constant $C > 0$ such that for any $\epsilon$ and a finite set of $n$ points, if $d \geq C\epsilon^{-2} \log(NV) \text{polylog}(n)$, the mapping $\boldsymbol{u}_t^i \mapsto \boldsymbol{v}_t^i$ satisfies the Johnson-Lindenstrauss lemma with high probability. Then, for all $i, j$,*

$$(1 - \epsilon)\|\boldsymbol{u}_t^i - \boldsymbol{u}_t^j\|_2^2 \leq \|\boldsymbol{v}_t^i - \boldsymbol{v}_t^j\|_2^2 \leq (1 + \epsilon)\|\boldsymbol{u}_t^i - \boldsymbol{u}_t^j\|_2^2.$$

---

### 3.3 SELECTION CRITERION

After computing the intra- and inter-sample importance scores $s_{\text{intra}}^{t,i}$ and $s_{\text{inter}}^{t,i}$, we combine them to obtain a joint score for each sample:

$$s_{\text{total}}^{t,i} = s_{\text{intra}}^{t,i} + \alpha s_{\text{inter}}^{t,i}, \tag{11}$$

where $\alpha$ is a trade-off factor. Based on $s_{\text{total}}^{t,i}$, we then select the top-$K$ samples from the current batch $\mathcal{B}_t$ for training:

$$\widehat{\mathcal{B}}_t = \arg \max_{S \subseteq \mathcal{B}_t, |S|=K} \sum_{i \in S} s_{\text{total}}^{t,i}. \tag{12}$$

The resulting subset selection module is plug-and-play, directly integrating with the SFT pipeline to enhance its performance while maintaining efficiency.

Table 3: **Performance Comparison for Different Online Batch Selection Methods.** We evaluate various methods across four benchmarks: MMLU, ScienceQA, GSM8K, and HumanEval. $\bar{A}$ denotes average accuracy or Pass@1 reported in percentage (%). $\mathcal{T}_{train}$ denotes throughput during the training time (samples per second). The best results of $\bar{A}$ are highlighted in **bold**. Methods training faster than the full dataset are marked in ▨, others are marked in ▨.

| Model | Method | MMLU | | ScienceQA | | GSM8K | | HumanEval | |
|---|---|---|---|---|---|---|---|---|---|
| | | $\bar{A}(\uparrow)$ | $\mathcal{T}_{train}(\uparrow)$ | $\bar{A}(\uparrow)$ | $\mathcal{T}_{train}(\uparrow)$ | $\bar{A}(\uparrow)$ | $\mathcal{T}_{train}(\uparrow)$ | $\bar{A}(\uparrow)$ | $\mathcal{T}_{train}(\uparrow)$ |
| Llama-3.1-8B | Regular | $38.24_{\pm0.35}$ | 2.09 | $93.19_{\pm0.65}$ | 6.61 | $55.95_{\pm0.47}$ | 3.73 | $29.28_{\pm0.48}$ | 5.77 |
| | Random | $35.47_{\pm1.25}$ | 3.74 | $92.86_{\pm0.27}$ | 10.06 | $54.89_{\pm0.73}$ | 6.68 | $26.74_{\pm0.56}$ | 8.92 |
| | MaxLoss | $35.62_{\pm0.79}$ | 2.75 | $92.82_{\pm0.21}$ | 7.49 | $55.42_{\pm0.35}$ | 4.98 | $27.23_{\pm0.27}$ | 6.62 |
| | MaxGrad | $35.91_{\pm1.17}$ | 0.29 | $92.77_{\pm0.24}$ | 0.94 | $55.08_{\pm0.58}$ | 0.53 | $26.89_{\pm0.74}$ | 0.80 |
| | RHO-Loss | $37.63_{\pm0.67}$ | 1.40 | $93.42_{\pm0.15}$ | 3.83 | $56.54_{\pm0.52}$ | 2.53 | $27.18_{\pm0.29}$ | 3.36 |
| | GREATS | $39.04_{\pm0.29}$ | 1.88 | $93.68_{\pm0.19}$ | 6.04 | $57.03_{\pm0.33}$ | 3.37 | $28.56_{\pm0.34}$ | 5.25 |
| | **UDS (Ours)** | $\mathbf{40.16_{\pm0.58}}$ | 2.48 | $\mathbf{94.33_{\pm0.28}}$ | 7.46 | $\mathbf{58.98_{\pm0.24}}$ | 4.59 | $\mathbf{30.96_{\pm0.69}}$ | 6.41 |
| Qwen-2.5-7B | Regular | $55.32_{\pm0.79}$ | 2.27 | $94.56_{\pm0.17}$ | 7.05 | $78.23_{\pm0.07}$ | 3.77 | $45.82_{\pm0.41}$ | 6.24 |
| | Random | $54.26_{\pm1.85}$ | 9.29 | $93.28_{\pm0.35}$ | 10.27 | $77.69_{\pm0.29}$ | 9.18 | $40.20_{\pm1.18}$ | 9.35 |
| | MaxLoss | $54.51_{\pm1.37}$ | 5.93 | $93.05_{\pm0.40}$ | 7.93 | $77.78_{\pm0.36}$ | 5.45 | $41.34_{\pm0.83}$ | 6.92 |
| | MaxGrad | $54.33_{\pm0.69}$ | 0.31 | $93.86_{\pm0.51}$ | 0.98 | $77.62_{\pm0.28}$ | 0.53 | $40.83_{\pm0.41}$ | 0.88 |
| | RHO-Loss | $57.08_{\pm0.74}$ | 1.94 | $93.80_{\pm0.25}$ | 3.89 | $78.38_{\pm0.43}$ | 3.08 | $43.08_{\pm1.62}$ | 3.53 |
| | GREATS | $58.19_{\pm0.49}$ | 2.12 | $94.17_{\pm0.62}$ | 6.53 | $78.61_{\pm0.41}$ | 3.50 | $45.04_{\pm0.59}$ | 5.80 |
| | **UDS (Ours)** | $\mathbf{63.34_{\pm0.36}}$ | 3.41 | $\mathbf{95.19_{\pm0.22}}$ | 7.85 | $\mathbf{79.91_{\pm0.23}}$ | 4.99 | $\mathbf{46.28_{\pm0.35}}$ | 6.81 |

# 4 EXPERIMENTS

## 4.1 EXPERIMENTAL SETUP

**Datasets and Backbones:** We evaluate online batch selection methods' performance across four key domains: (1) *general knowledge understanding* using the MMLU (Hendrycks et al., 2021a) benchmark, with auxiliary training datasets for fine-tuning and the official test set for evaluation; (2) *scientific question answering* using ScienceQA (Lu et al., 2022) for both fine-tuning and evaluation; (3) *mathematical reasoning* on GSM8K (Cobbe et al., 2021) for both fine-tuning and evaluation; and (4) *code generation* using CodeAlpaca-20k (Chaudhary, 2023) for training and HumanEval (Chen et al., 2021) for evaluation. All experiments are conducted using Llama-3.1-8B (Grattafiori et al., 2024) and Qwen-2.5-7B (Yang et al., 2024) as backbone models.

**Baselines:** We compare UDS against the following baselines: (1) **Regular**, where all samples are used without data selection; (2) **Random Selection**, which randomly chooses samples from batches; (3) **MaxLoss** (Loshchilov & Hutter, 2015), which prioritizes samples with the highest training loss; (4) **MaxGrad** (Katharopoulos & Fleuret, 2018), which selects samples with the largest gradient norms; (5) **RHO-Loss** (Mindermann et al., 2022), a representative method using a reference model; and (6) **GREATS** (Wang et al., 2024), a state-of-the-art (SOTA) online batch selection approach. All baselines select the same fraction of data for a fair comparison.

**Implementation Details:** Under hardware constraints, we employ LoRA (rank=8) for SFT training. The batch size is set to $B = 8$ for all datasets. The optimal data selection ratio $\alpha$ is highly dependent on both the backbone model and the dataset; the ratios we adopt for different combinations of backbones and datasets are reported in Table 5. The default choice of the buffer size $M = 1024$, the down-projection dimension is $d_1 = 128, d_2 = 8$, and we use this across all the experiments. For detailed sensitivity analysis, please refer to Appendix B.1. We report average accuracy on MMLU, ScienceQA, and GSM8K, and Pass@1 for HumanEval. We also compare training speed by throughput relative to an NVIDIA GeForce RTX 3090 GPU, following Wang et al. (2024). All benchmarks use zero-shot evaluation and are repeated four times with different random seeds. More detailed experimental settings are provided in Appendix D.

Table 4: **Ablation study of UDS components across multiple benchmarks using Qwen-2.5-7B.** $\bar{A}$ denotes average accuracy or Pass@1 reported in percentage (%). We report $\bar{A}$ and relative improvement $\Delta$ over the baseline.

| Method | MMLU | | ScienceQA | | GSM8K | | HumanEval | |
|---|---|---|---|---|---|---|---|---|
| | $\bar{A}$ (%) | $\Delta$ | $\bar{A}$ (%) | $\Delta$ | $\bar{A}$ (%) | $\Delta$ | $\bar{A}$ (%) | $\Delta$ |
| Random (Baseline) | $54.26_{\pm1.85}$ | – | $93.28_{\pm0.35}$ | – | $77.69_{\pm0.29}$ | – | $40.20_{\pm1.18}$ | – |
| Only Nuclear Norm | $58.35_{\pm0.76}$ | +4.09 | $94.19_{\pm0.31}$ | +0.91 | $79.22_{\pm0.16}$ | +1.53 | $44.18_{\pm0.55}$ | +3.98 |
| Only Diversity Distance | $57.75_{\pm1.48}$ | +3.49 | $93.98_{\pm0.24}$ | +0.70 | $78.96_{\pm0.31}$ | +0.67 | $43.84_{\pm0.19}$ | +3.64 |
| **UDS (Full)** | $\mathbf{63.34_{\pm0.36}}$ | +9.08 | $\mathbf{95.19_{\pm0.22}}$ | +1.91 | $\mathbf{79.91_{\pm0.23}}$ | +2.22 | $\mathbf{46.28_{\pm0.35}}$ | +6.08 |

## 4.2 Highest Accuracy with Lower Training Time than Full Dataset

**UDS achieves the highest accuracy.** Across all four benchmarks, UDS consistently delivers the best accuracy among online batch selection methods. On MMLU, it achieves 63.34%, outperforming GREATS by +5.15% when using Qwen-2.5-7B. Similar gains appear on ScienceQA (95.19% vs. 94.17%), GSM8K (79.91% vs. 78.61%), and HumanEval (46.28% vs. 45.04%). The improvements hold across both Llama-3.1-8B and Qwen-2.5-7B, showing strong generalization. Overall, UDS surpasses simple heuristics (MaxLoss, MaxGrad) and advanced baselines (RHO-Loss, GREATS), achieving SOTA performance in the online batch selection setting.

**UDS is more efficient than training on the full dataset.** Beyond accuracy, UDS maintains competitive efficiency relative to training on the full dataset. On Qwen-2.5-7B, it achieves 3.41 samples/s on MMLU and 6.81 on HumanEval, both higher than those of full-dataset training (2.27 and 6.24). On Llama-3.1-8B, it also sustains higher throughput while yielding better accuracy. MaxLoss also trains quickly but provides only marginal accuracy gains, whereas MaxGrad slows training dramatically with no significant benefit. GREATS, though accurate, consistently runs slower than UDS. Thus, UDS achieves the best trade-off, combining high accuracy with efficiency.

## 4.3 Ablation Studies

**Both components of UDS contribute to performance.** We conduct ablation studies on the two components of UDS, as summarized in Table 4. Using only the *Nuclear Norm* consistently outperforms random selection, underscoring the importance of capturing intra-sample utility and diversity. Using only the *Diversity Distance* also improves performance over the baseline, as discouraging redundancy among selected samples enhances overall utility. Combined, the two components complement each other to achieve the best results across all benchmarks, highlighting the necessity of jointly modeling sample utility, intra-sample diversity, and inter-sample diversity.

## 4.4 Comparison across Different Data Scales

To assess the effectiveness of UDS under varying batch selection fractions, we conduct experiments as shown in Figure 5, where the horizontal axis denotes the number of selected samples $K$ per batch and the vertical axis reports average accuracy.

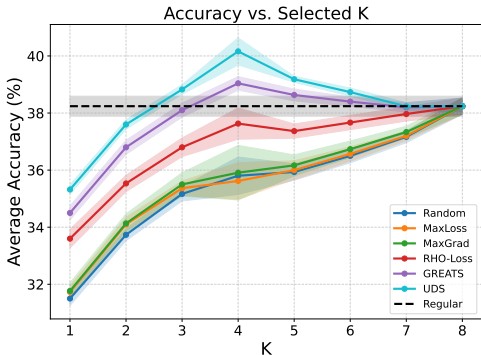

Figure 5: **Performance across different data scales using Llama-3.1-8B on MMLU.** We report accuracy when fine-tuning with varying proportions of training data. While all methods improve with more data, UDS consistently achieves the best accuracy and surpasses full-dataset fine-tuning.

While most methods improve with $K$, UDS's performance peaks at $K = 4$, achieving the highest accuracy before gradually declines as additional, less informative samples are added. When $K = B = 8$, all methods reduce to full-dataset fine-tuning. UDS consistently outperforms baselines,

showing its ability to prioritize informative and diverse samples. At its peak, UDS even surpasses full-dataset fine-tuning, demonstrating that a small, well-curated subset can deliver both efficiency and accuracy. These results confirm that UDS effectively balances exploitation and exploration, yielding robust improvements under different selection budgets.

## 5 CONCLUSION

In this work, we systematically analyze the limitations of current online batch selection methods for supervised fine-tuning of LLMs and establish three fundamental desiderata for optimal selection strategy design. To fulfill these requirements, we propose UDS, a novel framework that synergistically combines two complementary components: (i) Nuclear Norm scoring to capture both optimization utility and intra-sample diversity, and (ii) Diversity Distance measurement to ensure inter-sample diversity through efficient historical comparison. Extensive experiments across multiple domains illustrate UDS's superior performance over existing methods while maintaining computational efficiency. Our approach provides a competitive and practical solution for effective online batch selection and holds the potential of scaling to more efficient LLM training scenarios.

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

# A   RELATED WORK

**Online Batch Selection for Large Language Models.**   Several studies have investigated online batch selection prior to the era of LLMs. Works like (Jiang et al., 2019; Katharopoulos & Fleuret, 2018; Loshchilov & Hutter, 2015) propose heuristic methods that select "important" samples (e.g., via high training loss, or large gradient norm). These methods are straightforward and often effective for smaller models or simpler tasks, but more recent studies (Wang et al., 2024) show that these heuristics may not scale well for LLMs. Another paradigm uses a reference model to help evaluate sample importance. For instance, (Deng et al., 2023; Mindermann et al., 2022) use a pre-trained or separately trained reference model, and select samples based on how "different" they are relative to that model. These methods tend to require extra computation (e.g. extra forward passes) and sometimes held-out data to train the reference model, which can make them costly or impractical in large-scale LLM training. (Kaddour et al., 2023) points out that these overheads can outweigh benefits in many settings. GREATS (Wang et al., 2024) is a more recent SOTA method. It selects data batches online by considering loss reduction on a validation set, and uses a "ghost inner-product" approximation (via a Taylor-expansion-based greedy algorithm) to accelerate selection. However, the need for a validation set may be unavailable or non-representative in practical deployment (Wang et al., 2024).

**Offline Data Selection for Large Language Models.**   Most existing approaches to data selection are performed in an offline manner, where samples are filtered prior to training (Albalak et al., 2024; Xia et al., 2024; Zhou et al., 2023; Chen et al., 2023; Xie et al., 2023; Wettig et al., 2024). This is primarily due to efficiency concerns, since curating the entire dataset at every iteration is prohibitively expensive. Early approaches extend uncertainty estimation and active learning heuristics, such as filtering by entropy or difficulty, but these remain limited in capturing truly informative samples (Xu et al., 2020). A more impactful direction is large-scale pruning and deduplication: (Lee et al., 2021; Tirumala et al., 2023) show that removing near-duplicate or low-quality data improves efficiency and prevents overfitting. In instruction tuning, several works highlight that "quality outweighs quantity", with methods like LIMA (Zhou et al., 2023), LESS (Xia et al., 2024), and Self-Instruct (Wang et al., 2022) filtering diverse, representative subsets while discarding noisy prompts. Recent advances further explore offline data selection through various sampling strategies: (Sachdeva et al., 2024) introduce LLM-as-judge method and density-based sampling; (Axiotis et al., 2024) propose clustering-based sensitivity sampling; and (Deb et al., 2025) leverage information gain estimation to identify high-value examples. While effective to some extent, this perspective assumes that the value of each example is fixed throughout the learning process. In reality, learning is highly dynamic. Data that appear highly informative at the beginning of training may later become redundant (Wang et al., 2024). This limitation motivates serving online batch selection as a complement for dynamically assessing sample value during the training process.

# B   ADDITIONAL RESULTS

## B.1   PARAMETER ANALYSIS

**Impact of memory buffer size and low projected dimension.**   We evaluate the influence of the projected dimensions $d_1$, $d_2$, and the memory buffer size $M$ on both model performance and resource usage. As shown in Figures 6 and 7, increasing the projected dimensions and memory buffer consistently improves accuracy, which gradually saturates once $d_1$, $d_2$, and $M$ become sufficiently large. The trends of $d_1$ and $d_2$ are consistent with the Johnson-Lindenstrauss lemma (Johnson et al., 1984; Ailon & Chazelle, 2006; Jin et al., 2021; Ailon & Chazelle, 2009; Tropp, 2011), while a sufficiently large $M$ leads to performance saturation, indicating that the stored representations are adequate to capture global sample diversity. The corresponding increases in memory usage and computation time are minor, and arise only when computing inter-sample diversity.

In Figure 6, increasing $d_1$, $d_2$, and $M$ has no significant impact on extra memory usage. The base overhead of around 1.17GB comes from the projection matrices $\mathbf{\Gamma}_1$ and $\mathbf{\Gamma}_2$, which is small compared to the overall training memory footprint of about 22GB. In Figure 7, the additional time per batch is also negligible relative to the total training cost of 2.3s per batch.

Overall, the extra resource cost is minimal compared to the performance gains. In practice, parameter selection only needs to ensure that $d_1$, $d_2$, and $M$ are sufficiently large for stable performance.

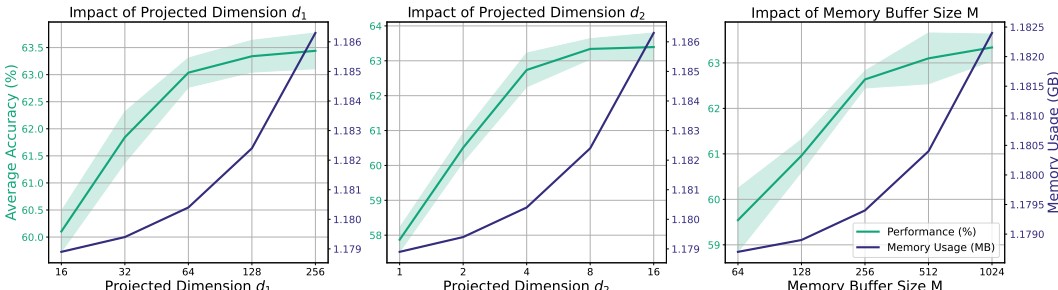

Figure 6: **Impact of low projected dimensions $d_1$ and $d_2$, and memory buffer size $M$ on model performance and additional memory usage for Qwen-2.5 on MMLU.** The green curves show average accuracy (%), and the blue curves show extra peak memory consumption (GB). Performance improves with increasing projected dimensions and memory buffer size, while additional memory usage increases only slightly.

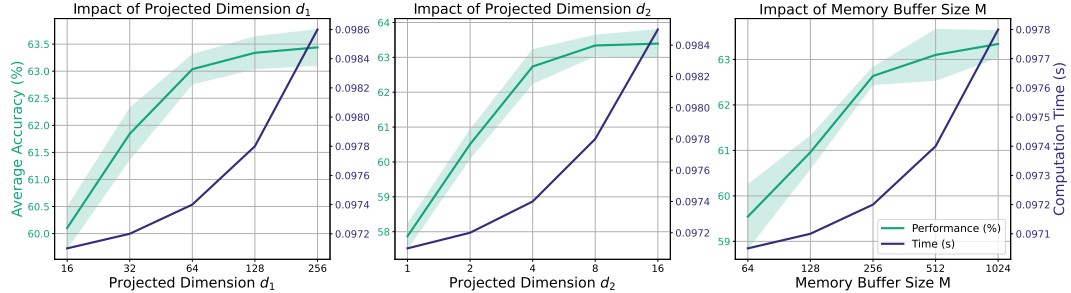

Figure 7: **Impact of low projected dimensions $d_1$ and $d_2$, and memory buffer size $M$ on model performance and additional computation time for Qwen-2.5 on MMLU.** The green curves show average accuracy (%), and the blue curves show additional computation time per batch (s). Performance improves with increasing projected dimensions and memory buffer size, while the additional computation time remains negligible.

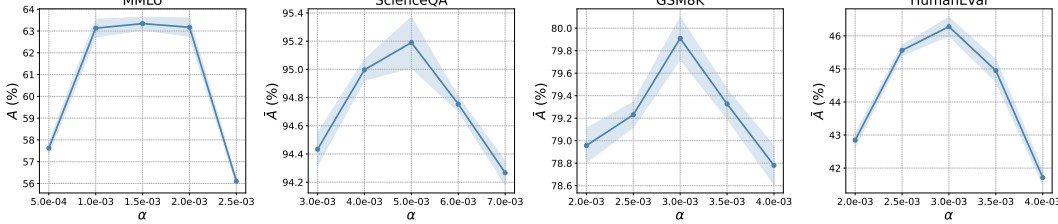

Figure 8: **Sensitivity analysis of the trade-off factor $\alpha$ on Qwen-2.5-7B.** Results are reported across four benchmarks—MMLU, ScienceQA, GSM8K, and HumanEval—using the data selection fractions specified in Table 5.

**Impact of trade-off factor $\alpha$.** Figure 8 presents the sensitivity analysis of $\alpha$ on Qwen-2.5-7B across four datasets. The results consistently exhibit an inverted U-shaped trend: performance improves as $\alpha$ increases, peaks at an optimal value, and then declines as $\alpha$ continues to grow. This pattern highlights the importance of maintaining a proper balance. The optimal $\alpha$ values corresponding to the selected data fractions in Table 5 are summarized in Table 6.

Table 5: **Data selection fractions across benchmark datasets.** Percentages of samples selected from MMLU, ScienceQA, GSM8K, and CodeAlpaca, evaluated with Llama-3.1-8B and Qwen-2.5-7B as backbone models.

| Model | MMLU | ScienceQA | GSM8K | CodeAlpaca |
|---|---|---|---|---|
| Llama-3.1-8B | 50% | 50% | 50% | 50% |
| Qwen-2.5-7B | 12.5% | 50% | 25% | 50% |

Table 6: **Optimal trade-off factor $\alpha$ across datasets and backbones.** Optimal $\alpha$ values for MMLU, ScienceQA, GSM8K, and CodeAlpaca using Llama-3.1-8B and Qwen-2.5-7B, with data selection fractions given in Table 5.

| Model | MMLU | ScienceQA | GSM8K | CodeAlpaca |
|---|---|---|---|---|
| Llama-3.1-8B | $4.5 \times 10^{-3}$ | $7 \times 10^{-3}$ | $1 \times 10^{-3}$ | $5 \times 10^{-3}$ |
| Qwen-2.5-7B | $1.5 \times 10^{-3}$ | $5 \times 10^{-3}$ | $3 \times 10^{-3}$ | $3 \times 10^{-3}$ |

### B.2 ADDITIONAL ABLATION STUDIES ON BUFFER UPDATE POLICIES, DISTANCE AGGREGATION STRATEGIES, AND RANDOM MATRIX CONSTRUCTIONS

For buffer update policies, we compare our default FIFO strategy with two alternatives: reservoir sampling (random replacement within the buffer) and class-aware sampling (constructing class prototypes via K-means). Experiments on the MMLU benchmark using Qwen-2.5-7B (Table 7) show that all three policies yield comparable performance, with no notable differences observed.

For distance aggregation strategies, we evaluate our average-distance scheme against two variants—farthest-$k$ and soft-min—again on MMLU with Qwen-2.5-7B (Table 8). The results similarly indicate no significant performance differences among these methods.

In UDS, we employ an SRFT-style construction for the bidirectional random projection matrix. To assess its necessity, we compare it against two alternative constructions: Sparse JL and CountSketch-style tensor sketches. Experiments on MMLU with Qwen-2.5-7B (Table 9) show that the CountSketch-style transform achieves performance nearly identical to the SRFT-style transform, whereas Sparse JL performs significantly worse.

The inferior performance of Sparse JL stems from its failure to satisfy the JL lemma under the bidirectional formulation, similar to the Gaussian-style construction discussed in Section 3.2. The issue arises because the Kronecker product disrupts the independence structure of the random variables, thereby violating the distance-preserving requirements of the JL lemma. In contrast, both SRFT-style and CountSketch-style constructions continue to satisfy the JL lemma when used bidirectionally.

From an efficiency perspective, decomposing the full random projection into a bidirectional one already greatly reduces memory usage and computation cost, making these overheads negligible relative to other components of the algorithm. Consequently, as long as a construction satisfies the JL lemma in the bidirectional setting, it is unnecessary to further distinguish which one is marginally more efficient. The SRFT-style construction is therefore sufficient for our purposes, and is what we adopt in our main experiments.

Table 7: **Ablation Studies on Buffer Update Policies.** We report average accuracy on MMLU using Qwen-2.5-7B.

| FIFO | Reservoir Sampling | Class-aware Sampling |
|---|---|---|
| $63.34_{\pm0.36}$ | $63.45_{\pm0.22}$ | $63.15_{\pm0.50}$ |

Table 8: **Ablation Studies on Distance Aggregation Policies.** We report average accuracy on MMLU using Qwen-2.5-7B.

| Average Distance | Farthest-k | Soft-min |
|---|---|---|
| $63.34_{\pm 0.36}$ | $63.18_{\pm 0.29}$ | $62.98_{\pm 0.48}$ |

Table 9: **Ablation Studies on Different Construction of the Random Matrix.** We report average accuracy on MMLU using Qwen-2.5-7B.

| SRFT-style | Sparse JL | CountSketch-style |
|---|---|---|
| $63.34_{\pm 0.36}$ | $55.28_{\pm 0.65}$ | $63.12_{\pm 0.31}$ |

### B.3    EXPERIMENTS ON FULL SFT, LARGER BATCH SIZES, AND INSTRUCTION-TUNED MODELS

To further demonstrate the scalability of UDS, we provide additional results under three settings: larger batch sizes (mini-batch size $B = 16$ in Table 10), full-parameter finetuning with Qwen-2.5-7B (Table 11), and instruction-tuned models using Qwen-2.5-7B-Instruct (Table 12). Across all scenarios, UDS consistently outperforms the corresponding baselines, strengthening the robustness and generalization capability of the method.

Table 10: **Accuracy Comparison for Different Online Batch Selection Methods with Larger Batch Size.** We evaluate various methods across four benchmarks: MMLU, ScienceQA, GSM8K, and HumanEval. $\bar{A}$ denotes average accuracy or Pass@1 reported in percentage (%). The best results of $\bar{A}$ are highlighted in **bold**.

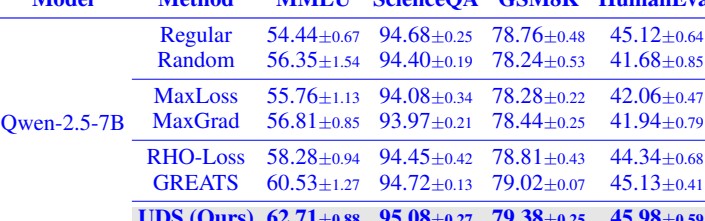

| Model | Method | MMLU | ScienceQA | GSM8K | HumanEval |
|---|---|---|---|---|---|
| Qwen-2.5-7B | Regular | $54.44_{\pm 0.67}$ | $94.68_{\pm 0.25}$ | $78.76_{\pm 0.48}$ | $45.12_{\pm 0.64}$ |
| | Random | $56.35_{\pm 1.54}$ | $94.40_{\pm 0.19}$ | $78.24_{\pm 0.53}$ | $41.68_{\pm 0.85}$ |
| | MaxLoss | $55.76_{\pm 1.13}$ | $94.08_{\pm 0.34}$ | $78.28_{\pm 0.22}$ | $42.06_{\pm 0.47}$ |
| | MaxGrad | $56.81_{\pm 0.85}$ | $93.97_{\pm 0.21}$ | $78.44_{\pm 0.25}$ | $41.94_{\pm 0.79}$ |
| | RHO-Loss | $58.28_{\pm 0.94}$ | $94.45_{\pm 0.42}$ | $78.81_{\pm 0.43}$ | $44.34_{\pm 0.68}$ |
| | GREATS | $60.53_{\pm 1.27}$ | $94.72_{\pm 0.13}$ | $79.02_{\pm 0.07}$ | $45.13_{\pm 0.41}$ |
| | **UDS (Ours)** | $\mathbf{62.71_{\pm 0.88}}$ | $\mathbf{95.08_{\pm 0.27}}$ | $\mathbf{79.38_{\pm 0.25}}$ | $\mathbf{45.98_{\pm 0.59}}$ |

### B.4    ADDITIONAL EXPERIMENTS ON TASKS REQUIRING LONG-CONTEXT REASONING

In all previous experiments, we set the maximum sequence length to 512. To assess whether UDS can effectively capture utility and diversity for training samples that require long-context reasoning, we further conduct experiments on the MATH dataset (Hendrycks et al., 2021b), using a maximum sequence length of 2048. We train on the MATH training split and evaluate on its test split. As shown in Table 13, UDS continues to deliver strong performance compared with other baselines, even under this more challenging long-context setting.

### B.5    ADDITIONAL EXPERIMENTS ON OOD EVALUATION

To further assess the robustness of UDS, we conduct out-of-distribution (OOD) evaluations using GSM8K as a case study. Specifically, we train on the GSM8K training set and evaluate on its in-distribution test set as well as two OOD benchmarks: MATH500 and AMC23 (Lightman et al., 2023). As shown in Table 14, UDS continues to exhibit strong performance across both in-distribution and OOD settings, demonstrating its robustness to distribution shifts.

### B.6    STRONG CORRELATION BETWEEN THE FROBENIUS NORM OF THE LOGITS MATRIX AND SAMPLE UTILITY

Table 11: **Accuracy Comparison for Different Online Batch Selection Methods with Full Fine-tuning.** We evaluate various methods across four benchmarks: MMLU, ScienceQA, GSM8K, and HumanEval. $\bar{A}$ denotes average accuracy or Pass@1 reported in percentage (%). The best results of $\bar{A}$ are highlighted in **bold**.

| Model | Method | MMLU | ScienceQA | GSM8K | HumanEval |
|---|---|---|---|---|---|
| | Regular | $55.64_{\pm0.43}$ | $94.52_{\pm0.12}$ | $77.56_{\pm0.16}$ | $48.03_{\pm0.33}$ |
| | Random | $56.05_{\pm0.71}$ | $94.38_{\pm0.25}$ | $76.95_{\pm0.35}$ | $41.46_{\pm0.56}$ |
| | MaxLoss | $55.59_{\pm0.27}$ | $94.41_{\pm0.19}$ | $76.82_{\pm0.42}$ | $42.36_{\pm0.60}$ |
| Qwen-2.5-7B | MaxGrad | $55.16_{\pm0.61}$ | $94.27_{\pm0.32}$ | $77.08_{\pm0.25}$ | $43.29_{\pm0.52}$ |
| | RHO-Loss | $57.94_{\pm1.73}$ | $94.64_{\pm0.21}$ | $77.51_{\pm0.36}$ | $46.09_{\pm0.40}$ |
| | GREATS | $58.86_{\pm0.54}$ | $94.75_{\pm0.23}$ | $77.86_{\pm0.17}$ | $47.56_{\pm0.49}$ |
| | **UDS (Ours)** | $\mathbf{63.27_{\pm0.33}}$ | $\mathbf{95.06_{\pm0.28}}$ | $\mathbf{78.26_{\pm0.39}}$ | $\mathbf{48.44_{\pm0.70}}$ |

Table 12: **Accuracy Comparison for Different Online Batch Selection Methods on Instruction Tuning Model.** We evaluate various methods across four benchmarks: MMLU, ScienceQA, GSM8K, and HumanEval. $\bar{A}$ denotes average accuracy or Pass@1 reported in percentage (%). The best results of $\bar{A}$ are highlighted in **bold**.

| Model | Method | MMLU | ScienceQA | GSM8K | HumanEval |
|---|---|---|---|---|---|
| | Regular | $50.15_{\pm0.15}$ | $94.47_{\pm0.18}$ | $75.74_{\pm0.19}$ | $45.29_{\pm0.37}$ |
| | Random | $47.38_{\pm1.41}$ | $94.60_{\pm0.32}$ | $75.36_{\pm0.29}$ | $41.09_{\pm0.64}$ |
| | MaxLoss | $47.92_{\pm0.87}$ | $94.92_{\pm0.53}$ | $75.57_{\pm0.42}$ | $41.17_{\pm0.41}$ |
| Qwen-2.5-7B-Instruct | MaxGrad | $47.63_{\pm0.51}$ | $94.46_{\pm0.36}$ | $75.63_{\pm0.51}$ | $41.24_{\pm0.38}$ |
| | RHO-Loss | $49.62_{\pm0.81}$ | $95.14_{\pm0.26}$ | $76.15_{\pm0.26}$ | $43.19_{\pm0.23}$ |
| | GREATS | $52.69_{\pm1.03}$ | $95.22_{\pm0.19}$ | $76.20_{\pm0.35}$ | $44.36_{\pm0.52}$ |
| | **UDS (Ours)** | $\mathbf{53.86_{\pm0.42}}$ | $\mathbf{95.56_{\pm0.14}}$ | $\mathbf{76.80_{\pm0.37}}$ | $\mathbf{45.75_{\pm0.28}}$ |

Similar to Figures 2 and 3, we also observe a strong empirical correlation between the Frobenius norm of the logits matrix, $\|\boldsymbol{L}(\boldsymbol{x}_t^i; \boldsymbol{\theta}_t)\|_F$, and the change in loss, $-\delta\ell(\boldsymbol{x}_t^i; \boldsymbol{\theta}_t)$, as illustrated in Figure 9.

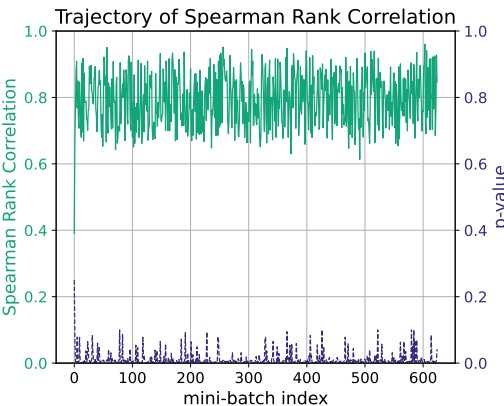

Figure 9: Strong correlation between $-\delta\ell(\boldsymbol{x}_t^i; \boldsymbol{\theta}_t)$ and $\|\boldsymbol{L}(\boldsymbol{x}_t^i; \boldsymbol{\theta}_t)\|_F$ during the training process. Each point represents the correlation coefficient calculated from $B$ pairs of values within a single batch. We extract the result using Qwen-2.5-7B on MMLU.

### B.7 COMPARISON WITH OFFLINE DATA SELECTION METHODS

We additionally include a representative offline data selection method, FisherSFT (Deb et al., 2025), which uses Fisher information gain to select a training subset, in our comparison. We conduct experiments on all four datasets using Qwen-2.5-7B as the backbone model. Since FisherSFT does not operate in the same manner as online batch selection methods, it is infeasible to fairly compare

Table 13: **Additional Studies on MATH dataset with Longer CoT Reasoning.** We report average accuracy using Qwen-2.5-7B.

| Regular | Random | MaxLoss | MaxGrad | RHO-Loss | GREATS | UDS (Ours) |
|---------|--------|---------|---------|----------|--------|------------|
| $45.89_{\pm 0.14}$ | $42.85_{\pm 0.28}$ | $42.69_{\pm 0.48}$ | $42.77_{\pm 0.61}$ | $45.35_{\pm 0.32}$ | $45.66_{\pm 0.29}$ | $\mathbf{46.27_{\pm 0.35}}$ |

Table 14: **Accuracy Comparison for Different Online Batch Selection Methods on OOD Datasets.** We evaluate various methods across three benchmarks: GSM8K (in-distribution), MATH500, and AMC23 (out-of-distribution). $\bar{A}$ denotes average accuracy or Pass@1 reported in percentage (%). The best results of $\bar{A}$ are highlighted in **bold**.

| Model | Method | GSM8K | MATH500 | AMC23 |
|-------|--------|-------|---------|-------|
| Qwen-2.5-7B | Regular | $78.23_{\pm 0.07}$ | $41.83_{\pm 0.38}$ | $24.67_{\pm 0.73}$ |
| | Random | $77.69_{\pm 0.29}$ | $41.09_{\pm 0.24}$ | $24.39_{\pm 0.64}$ |
| | MaxLoss | $77.78_{\pm 0.36}$ | $41.18_{\pm 0.34}$ | $23.89_{\pm 1.12}$ |
| | MaxGrad | $77.62_{\pm 0.28}$ | $41.13_{\pm 0.27}$ | $24.26_{\pm 0.58}$ |
| | RHO-Loss | $78.38_{\pm 0.43}$ | $41.84_{\pm 0.19}$ | $25.59_{\pm 0.69}$ |
| | GREATS | $78.61_{\pm 0.41}$ | $42.27_{\pm 0.39}$ | $25.97_{\pm 0.76}$ |
| | **UDS (Ours)** | $\mathbf{79.91_{\pm 0.23}}$ | $\mathbf{42.56_{\pm 0.74}}$ | $\mathbf{26.38_{\pm 0.83}}$ |

throughput, so we primarily compare the final accuracy after training. Both FisherSFT and UDS select the same fraction of data, and all other training details remain the same. The results in Table 15 show that UDS also achieves better performance than FisherSFT.

Table 15: **Accuracy Comparison with Offline Data Selection Strategy FisherSFT.** We report average accuracy using Qwen-2.5-7B.

| Method | MMLU | ScienceQA | GSM8K | HumanEval |
|--------|------|-----------|-------|-----------|
| **FisherSFT** | $57.85_{\pm 0.62}$ | $94.02_{\pm 0.34}$ | $78.35_{\pm 0.30}$ | $43.87_{\pm 0.59}$ |
| **UDS (Ours)** | $\mathbf{63.34_{\pm 0.36}}$ | $\mathbf{95.19_{\pm 0.22}}$ | $\mathbf{79.91_{\pm 0.23}}$ | $\mathbf{46.28_{\pm 0.35}}$ |

## C  THEORETICAL ANALYSIS

### C.1  RELATIONSHIP BETWEEN FROBENIUS NORM AND NUCLEAR NORM

Lemma 3.1 establishes a fundamental relationship between two widely used matrix norms: the Frobenius norm $\|\cdot\|_F$ and the nuclear norm $\|\cdot\|_*$. Recall that if $\boldsymbol{L}(\boldsymbol{x}_t^i; \boldsymbol{\theta}_t) \in \mathbb{R}^{N \times V}$ has singular values $\{\sigma_j\}_{j=1}^r$ with rank $r \leq \min(N, V)$, then

$$\|\boldsymbol{L}(\boldsymbol{x}_t^i; \boldsymbol{\theta}_t)\|_F = \Big(\sum_{j=1}^r \sigma_j^2\Big)^{1/2}, \quad \|\boldsymbol{L}(\boldsymbol{x}_t^i; \boldsymbol{\theta}_t)\|_* = \sum_{j=1}^r \sigma_j. \tag{13}$$

The two-sided inequality in Lemma 3.1 is proved as follows.

(1) *Lower bound.* Since the Euclidean norm of a set of nonnegative numbers is always no larger than their sum, we have

$$\|\boldsymbol{L}(\boldsymbol{x}_t^i; \boldsymbol{\theta}_t)\|_F = \Big(\sum_{j=1}^r \sigma_j^2\Big)^{1/2} \leq \sum_{j=1}^r \sigma_j = \|\boldsymbol{L}(\boldsymbol{x}_t^i; \boldsymbol{\theta}_t)\|_*. \tag{14}$$

Equality holds iff at most one singular value is nonzero, i.e., $\boldsymbol{L}(\boldsymbol{x}_t^i; \boldsymbol{\theta}_t)$ is rank-1 (in which case the Euclidean norm and the sum of singular values coincide).

(2) *Upper bound.* Applying the Cauchy–Schwarz inequality to the vectors $(1, \ldots, 1) \in \mathbb{R}^r$ and $(\sigma_1, \ldots, \sigma_r) \in \mathbb{R}^r$ yields

$$\|\boldsymbol{L}(\boldsymbol{x}_t^i; \boldsymbol{\theta}_t)\|_* = \sum_{j=1}^{r} \sigma_j \leq \Big(\sum_{j=1}^{r} 1^2\Big)^{1/2} \Big(\sum_{j=1}^{r} \sigma_j^2\Big)^{1/2} = \sqrt{r}\|\boldsymbol{L}(\boldsymbol{x}_t^i; \boldsymbol{\theta}_t)\|_F. \tag{15}$$

Since $r \leq \min(N, V)$ we obtain the stated upper bound

$$\|\boldsymbol{L}(\boldsymbol{x}_t^i; \boldsymbol{\theta}_t)\|_* \leq \sqrt{\min(N, V)}\|\boldsymbol{L}(\boldsymbol{x}_t^i; \boldsymbol{\theta}_t)\|_F. \tag{16}$$

Equality in the Cauchy–Schwarz step holds iff the singular-value vector is proportional to the all-ones vector, i.e. all nonzero singular values are equal. Thus the rightmost equality is achieved when $r = \min(N, V)$ (full possible rank) and the $r$ singular values are equal.

**Implications.** This lemma highlights that the nuclear norm and Frobenius norm, though correlated, capture different structural aspects of $\boldsymbol{L}(\boldsymbol{x}_t^i; \boldsymbol{\theta}_t)$. The Frobenius norm measures the overall magnitude of logits, while the nuclear norm is also sensitive to their rank structure. A large nuclear norm relative to the Frobenius norm implies more "spread-out" singular values and thus higher intra-sample diversity.

## C.2 Optimization Analysis using Adam

When using the Adam (Kingma & Ba, 2014) as optimizer, the parameter update at step $t$ is

$$\boldsymbol{\theta}_{t+1} - \boldsymbol{\theta}_t = -\eta_t \cdot \frac{\hat{\boldsymbol{m}}_t}{\sqrt{\hat{\boldsymbol{v}}_t} + \epsilon_{\text{adam}}}, \tag{17}$$

where $\hat{\boldsymbol{m}}_t$ and $\hat{\boldsymbol{v}}_t$ are the bias-corrected first and second moment estimates of the stochastic gradients, respectively:

$$\hat{\boldsymbol{m}}_t = \frac{\boldsymbol{m}_t}{1 - \beta_1^t}, \quad \hat{\boldsymbol{v}}_t = \frac{\boldsymbol{v}_t}{1 - \beta_2^t}, \quad \boldsymbol{m}_t = \beta_1 \boldsymbol{m}_{t-1} + (1 - \beta_1) \cdot \boldsymbol{g}_t, \quad \boldsymbol{v}_t = \beta_2 \boldsymbol{v}_{t-1} + (1 - \beta_2) \cdot \boldsymbol{g}_t^2, \tag{18}$$

with $\boldsymbol{g}_t = \frac{1}{B} \sum_{i=1}^{B} \nabla_{\boldsymbol{\theta}} \ell(\boldsymbol{x}_t^i; \boldsymbol{\theta}_t)$.

After the parameter update $\boldsymbol{\theta}_t \to \boldsymbol{\theta}_{t+1}$, the logits matrix for datapoint $\boldsymbol{x}_t^i$ can be expanded using a first-order Taylor expansion:

$$\boldsymbol{L}(\boldsymbol{x}_t^i; \boldsymbol{\theta}_{t+1}) \approx \boldsymbol{L}(\boldsymbol{x}_t^i; \boldsymbol{\theta}_t) + \nabla_{\boldsymbol{\theta}} \boldsymbol{L}(\boldsymbol{x}; \boldsymbol{\theta}_t) \cdot (\boldsymbol{\theta}_{t+1} - \boldsymbol{\theta}_t), \tag{19}$$

so that the induced change is

$$\delta \boldsymbol{L}(\boldsymbol{x}; \boldsymbol{\theta}_t) \approx -\eta_t \, \nabla_{\boldsymbol{\theta}} \boldsymbol{L}(\boldsymbol{x}; \boldsymbol{\theta}_t) \cdot \frac{\hat{\boldsymbol{m}}_t}{\sqrt{\hat{\boldsymbol{v}}_t} + \epsilon_{\text{adam}}}. \tag{20}$$

## C.3 Distance Preservation under Low-dimensional Projection

We show that our two-sided projection construction can be algebraically reduced to a single subsampled randomized Fourier transform (SRFT) on $\mathbb{R}^{NV}$. Recall the standard property of the vec operator with Kronecker products: for conformable matrices $\boldsymbol{A}, \boldsymbol{B}, \boldsymbol{X}$, we have

$$\text{vec}(\boldsymbol{A}\boldsymbol{X}\boldsymbol{B}) = (\boldsymbol{B}^\top \otimes \boldsymbol{A}) \, \text{vec}(\boldsymbol{X}). \tag{21}$$

Applying this to $\boldsymbol{\Gamma}_2 \in \mathbb{R}^{d_2 \times N}$, $\boldsymbol{\Gamma}_1 \in \mathbb{R}^{d_1 \times V}$, and $\boldsymbol{L}(\boldsymbol{x}_t^i; \boldsymbol{\theta}_t) \in \mathbb{R}^{N \times V}$, we obtain

$$\boldsymbol{z}_t^i = \text{vec}(\boldsymbol{\Gamma}_2 \boldsymbol{L}(\boldsymbol{x}_t^i; \boldsymbol{\theta}_t)\boldsymbol{\Gamma}_1^\top) = (\boldsymbol{\Gamma}_1 \otimes \boldsymbol{\Gamma}_2) \, \text{vec}(\boldsymbol{L}(\boldsymbol{x}_t^i; \boldsymbol{\theta}_t)) = \boldsymbol{\Gamma} \, \text{vec}(\boldsymbol{L}(\boldsymbol{x}_t^i; \boldsymbol{\theta}_t)), \tag{22}$$

where $\boldsymbol{\Gamma} = \boldsymbol{\Gamma}_1 \otimes \boldsymbol{\Gamma}_2 \in \mathbb{R}^{d_1 d_2 \times NV}$. Using the Kronecker identity $(\boldsymbol{A} \otimes \boldsymbol{B})(\boldsymbol{C} \otimes \boldsymbol{D}) = (\boldsymbol{A}\boldsymbol{C}) \otimes (\boldsymbol{B}\boldsymbol{D})$, we can rewrite

$$\boldsymbol{\Gamma} = (\sqrt{\frac{V}{d_1}} \boldsymbol{S}_1 \boldsymbol{F}_1 \boldsymbol{D}_1) \otimes (\sqrt{\frac{N}{d_2}} \boldsymbol{S}_2 \boldsymbol{F}_2 \boldsymbol{D}_2) \tag{23}$$

$$= \sqrt{\frac{NV}{d_1 d_2}} (\boldsymbol{S}_1 \otimes \boldsymbol{S}_2)(\boldsymbol{F}_1 \otimes \boldsymbol{F}_2)(\boldsymbol{D}_1 \otimes \boldsymbol{D}_2), \tag{24}$$

where $\boldsymbol{S} = \boldsymbol{S}_1 \otimes \boldsymbol{S}_2$, $\boldsymbol{F} = \boldsymbol{F}_1 \otimes \boldsymbol{F}_2$, and $\boldsymbol{D} = \boldsymbol{D}_1 \otimes \boldsymbol{D}_2$. Since each $\boldsymbol{F}_i$ is orthonormal and $\boldsymbol{D}_i$ has independent Rademacher entries, $\boldsymbol{F}$ is orthonormal and $\boldsymbol{D}$ remains a diagonal matrix with independent Rademacher entries. Hence, $\boldsymbol{\Gamma} = \sqrt{\frac{NV}{d_1 d_2}} \boldsymbol{SFD}$ is exactly an SRFT.

**Remark:** This Kronecker structure preserves the SRFT type. In contrast, if $\boldsymbol{\Gamma}_1, \boldsymbol{\Gamma}_2$ are i.i.d. Gaussian (as in classical JL), then $\boldsymbol{\Gamma}_1 \otimes \boldsymbol{\Gamma}_2$ no longer has i.i.d. Gaussian entries due to induced correlations.

**Johnson-Lindenstrauss Guarantee.** Let $\boldsymbol{u}_1, \ldots, \boldsymbol{u}_n \in \mathbb{R}^{NV}$ denote $n$ vectors of interest. Consider all pairwise differences $\boldsymbol{u}_i - \boldsymbol{u}_j$, $1 \le i < j \le n$; there are at most $n(n-1)/2$ such vectors. By standard results on Fast JL transforms (SRFT/FJLT) (Ailon & Chazelle, 2006; Jin et al., 2021; Ailon & Chazelle, 2009; Tropp, 2011), for any $0 < \epsilon < 1$ and a fixed set of $n$ vectors in $\mathbb{R}^{NV}$, a random SRFT $\boldsymbol{\Gamma} \in \mathbb{R}^{d_1 d_2 \times NV}$ preserves the $\ell_2$ norms up to $(1 \pm \epsilon)$ with high probability provided

$$d_1 d_2 \gtrsim C \epsilon^{-2} \log(NV) \operatorname{polylog}(n), \tag{25}$$

where $C$ is an absolute constant and the polylog factor depends on the SRFT construction. This gives that, with high probability, for all $i, j$:

$$(1 - \epsilon) \|\boldsymbol{u}_i - \boldsymbol{u}_j\|_2^2 \le \|\boldsymbol{\Gamma}(\boldsymbol{u}_i - \boldsymbol{u}_j)\|_2^2 \le (1 + \epsilon) \|\boldsymbol{u}_i - \boldsymbol{u}_j\|_2^2, \tag{26}$$

which is exactly the claimed Johnson-Lindenstrauss distance preservation.

# D  DETAILED EXPERIMENTAL SETTING

## D.1  DATASETS

We conduct our experiments on the following datasets, with important information listed in Table 16. Brief introduction for each dataset are provided in the following:

- **MMLU** Hendrycks et al. (2021a) is a benchmark designed to assess general knowledge understanding, consisting of multiple-choice questions from various branches of knowledge. It covers 57 tasks including elementary mathematics, US history, computer science, law, and more.

- **ScienceQA** Lu et al. (2022) is a multimodal science question answering dataset collected from elementary and high school science curricula, including multiple-choice problems that align with California Common Core Content Standards. The questions in dataset are sourced from open resources managed by IXL Learning, an online learning platform curated by experts in the field of K-12 education. We extract textual parts within it following Li et al. (2024).

- **GSM8K** Cobbe et al. (2021) is a dataset of 8.5K high quality linguistically diverse grade school math word problems, each requiring multi-step arithmetic reasoning. Answers are provided in a step-by-step explanation format.

- **CodeAlpaca-20k** Chaudhary (2023) is an instruction-following dataset consisting of 20k examples generated to align code-related prompts with helpful outputs. It is synthetically constructed to enhance code instruction tuning.

- **HumanEval** Chen et al. (2021) is a code generation benchmark including 164 Python programming problems. The dataset was handwritten to ensure not to be included in the training set of code generation models.

## D.2  TRAINING CONFIGURATION

In this section, we detail the experiment configuration with hyper parameters. General configurations across all backbones and datasets are listed in Table 17. Dataset-specific and model-specific setting are listed in Table 18. For all baseline approaches, we adopt the same configurations described above. For GREATS, we set the validation set size to 5. For RHO-Loss, we use Llama-2-13B as the reference model for Llama-3.1-8B, and Qwen-2.5-14B as the reference model for Qwen-2.5-7B. Ideally, Llama-3.1-70B would serve as a better reference model for Llama-3.1-8B, but its computational cost makes it impractical in our setting. Fortunately, we find that Llama-2-13B provides sufficiently strong guidance, and thus we adopt it as a surrogate reference model.

Table 16: **Details of MMLU, ScienceQA, GSM8K, CodeAlpaca and HumanEval Datasets.** We list the number of training and testing samples and task types for the following datasets used in our experiments.

| Dataset | Training Samples | Testing Samples | Task Types |
|---|---|---|---|
| MMLU Hendrycks et al. (2021a) | 99842 | 14042 | Multiple Choice |
| ScienceQA Lu et al. (2022) | 12726 | 4241 | Multiple Choice |
| GSM8K Cobbe et al. (2021) | 7473 | 1319 | Math Problems |
| CodeAlpaca-20k Chaudhary (2023) | 20022 | — | Code Instruction |
| HumanEval Chen et al. (2021) | — | 164 | Code Generation |

Table 17: **General Training Hyperparameters with LoRA.** Shared configuration across all experiments, including rank settings, optimizer details, and architectural choices.

| Parameter | Value |
|---|---|
| Rank ($r$) | 8 |
| Scaling factor ($\alpha$) | 16 |
| Target modules | $\{$q,k,v,o,gate,down,up$\}$_proj |
| Optimizer | AdamW |
| Warmup ratio | 0.01 |
| Gradient accumulated batch | 128 |
| Dropout rate | 0.00 |

# E    LIMITATIONS AND FUTURE WORK

**Limitations.**    One potential limitation of our framework lies in the computation of the nuclear norm. While it can be obtained directly from the singular values of the logits matrix without incurring additional expensive backpropagation, the SVD of large matrices may still introduce non-negligible overhead in practice, particularly for long sequences with large vocabulary sizes. We also explored approximate methods such as randomized SVD to accelerate the computation, but observed that the loss in precision often degrades the overall performance. Another limitation is the difficulty of providing a rigorous theoretical proof for the linear correlation between the nuclear norm of the logits matrix and the resulting loss reduction or effective matrix rank.

**Future Work.**    An important future direction is to explore more efficient and accurate estimators of the nuclear norm. Potential avenues include exploiting structured low-rank approximations, leveraging iterative or block-wise SVD techniques, or designing surrogate scoring functions that preserve the optimization and diversity signals of the nuclear norm while being cheaper to compute. Such improvements would further enhance the scalability of our framework to larger models and longer input sequences.

# F    LLM USAGE DECLARATION

In preparing this manuscript, we used a large language model solely as a language assistance tool. Specifically, the LLM was employed to polish the phrasing of certain paragraphs and to improve clarity and readability of the text. All technical ideas, derivations, experiments, and analyses were conceived, implemented, and validated entirely by the authors. The LLM was not used for generating research ideas, designing experiments, or producing novel scientific content. The authors take full responsibility for all contents of the paper.

Table 18: **Dataset-Specific and Model-Specific Training Configurations during training.** Task-optimized settings for Llama-3.1-8B and Qwen-2.5-7B across four benchmarks, showing variations in epoch counts, learning rates, and sequence lengths based on dataset characteristics and model requirements.

| Model | Parameter | MMLU | ScienceQA | GSM8K | CodeAlpaca |
|---|---|---|---|---|---|
| Llama-3.1-8B | Epochs | 1 | 20 | 1 | 2 |
| | Learning rate | $1.5 \times 10^{-4}$ | $3 \times 10^{-4}$ | $3 \times 10^{-4}$ | $3 \times 10^{-4}$ |
| | Max sequence length | 512 | 256 | 512 | 512 |
| | micro batch size | 8 | 8 | 8 | 8 |
| Qwen-2.5-7B | Epochs | 1 | 20 | 1 | 2 |
| | Learning rate | $1.5 \times 10^{-4}$ | $3 \times 10^{-4}$ | $3 \times 10^{-4}$ | $3 \times 10^{-4}$ |
| | Max sequence length | 512 | 256 | 512 | 512 |
| | micro batch size | 8 | 8 | 8 | 8 |

