# OpenReview forum: "Utility-Diversity Aware Online Batch Selection for LLM Supervised Fine-tuning"
_ICLR.cc/2026/Conference — Submitted to ICLR 2026_

### Official Review · Reviewer_zKZs · 2025-10-31

**Soundness:** 2
**Presentation:** 2
**Contribution:** 2
**Rating:** 2
**Confidence:** 4

**Summary:**

This paper introduced a novel online batch selection framework for supervised fine-tuning in LLMs. The crucial idea of this framework was to design the sample importance score by taking utility and diversity into consideration. Experimental results showed that the proposed framework provided improved fine-tuning performance and reduced training time.

**Strengths:**

**Originality:** This work presented a novel online batch selection framework by jointly considering data utility, intra-sample diversity, and inter-sample diversity. It connected the nuclear norm of the logits matrix with both data utility and intra-sample diversity. It designed a structured bilinear random projection of logits for compact embedding when estimating the inter-sample diversity.

**Quality:** The correlation between the loss reduction and the nuclear norm of the logits matrix was empirically justified. Experiments demonstrated the superior performance of the proposed online batch selection framework over baselines. The ablation studies also confirmed the impact of the data utility/intra-sample diversity and inter-sample diversity components.

**Clarity:** The notions of data utility, intra-sample diversity, and inter-sample diversity were clearly presented. The key theoretical analysis was highlighted.

**Significance:** The proposed framework shows the impact of data selection in supervised fine-tuning in LLMs. It would largely advance the data valuation areas in LLMs by automatically selecting valuable samples.

**Weaknesses:**

(1) The theoretical analysis of the intra-sample importance score via the nuclear norm is unconvincing.
- It is unclear why the nuclear norm is used, instead of the Frobenius norm discussed in Lemma 3.1. Given the connections of these two norms, what are the unique advantages of the nuclear norm to characterize the intra-sample importance score?
- It is highlighted that **High Nuclear Norm Indicates High rank and orthogonal rows (high diversity)**. It is confusing why the high nuclear norm can guarantee the high rank of the logits matrix. Here is a possible counterexample: there are two matrices $A = [[10, 0], [ 0, 0]]$ and $A = [[1, 0], [ 0, 1]]$, it holds that $rank(A) =1< rank(B)=2$, but $\Vert A \Vert_*=10 > \Vert B \Vert_*=2$.
- The analysis for the connection between the nuclear norm and the optimization utility is also unclear. It is confusing how Eq. (8) provides insights into understanding the correlation between the nuclear/Frobenius norm and the optimization utility. Their correlation is only empirically analyzed in Figure 2. It is unclear whether this correlation holds theoretically.

(2) It seems that the diversity distance of Eq. (9) focuses only on the distinction from recent data $Q$. It doesn't measure the inter-sample diversity of samples in the current batch $\mathcal{B}_t$. If so, it is possible to choose repeated samples from $\mathcal{B}_t$. Besides, a larger buffer size $M$ might lead to better measurement of the inter-sample diversity, but will also produce high memory requirements. It would be better to show the impact of the value of $M$ in balancing the inter-sample diversity measurement and the memory requirement.

(3) Some experimental settings are missing, e.g., the value of the buffer size $M$, the trade-off factor $\alpha$, etc.

**Questions:**

(1) Why does it require considering the optional history $\mathcal{H}_t$ in the optimization problem of Eq. (1)?

(2) How would the discrete Fourier transform (DFT) matrices $F_1, F_2$ be selected?

(3) What is the value of the trade-off factor $\alpha$ in Eq. (11)? How will it affect the online batch selection performance?

(4) What is the value of $M$ in the experiments?

(5) How is the representation $z$ obtained in subsection 3.2, given the autoregressive language models in subsection 2.1?

---

> ### Author Response · Authors · 2025-11-27
> **Official Response to Reviewer zKZs from Author (Part I)**
>
> We thank the reviewer for the constructive feedback and provide detailed responses below. We have also included additional experiments and analyses in the revised manuscript.
>
> > It is highlighted that High Nuclear Norm Indicates High rank and orthogonal rows (high diversity). It is confusing why the high nuclear norm can guarantee the high rank of the logits matrix. Here is a possible counterexample: there are two matrices $A=[[10,0], [0,0]]$ and $A=[[1,0], [0,1]]$, it holds that $rank(A)=1<rank(B)=2$, but $||A||\*=10>||B||\*=2$.
>
> We apologize for the potential misunderstanding. We think that the verb "indicates" may not be sufficiently precise. What we truly intend to express is that "a high nuclear norm **positively correlates** with high rank and orthogonal rows (i.e., high intra-sample diversity)". We have revised the expression in the manuscript. We do not expect that a higher nuclear norm **always** means a higher rank,  but this is a phenomenon that **usually occurs** in practice.
>
> To justify this, we provide an additional experiment in **Figure 3** evaluating the Spearman rank correlation between the logits matrix's nuclear norm and its rank during training, similar to the experiment in Figure 2. According to Figure 3, we observe a strong and stable correlation (Spearman ≈ 0.8 throughout training). This phenomenon has also been recognized in prior work [1-2], where nuclear norm is used to approximate matrix diversity.
>
> > It is unclear why the nuclear norm is used, instead of the Frobenius norm discussed in Lemma 3.1. Given the connections of these two norms, what are the unique advantages of the nuclear norm to characterize the intra-sample importance score?
>
> We use the nuclear norm because the Frobenius norm only captures loss reduction (we additionally provide a Spearman correlation experiment between the logits matrix's Frobenius norm and loss reduction in **Figure 9** to strengthen this (Spearman ≈ 0.8 throughout training)), whereas the nuclear norm additionally captures intra-sample diversity; both properties benefit model learning. As we clarified in lines 198–201, since the bound in Lemma 3.1 is typically loose, with a factor of $\sqrt{\min(N,V)}$ between the lower and upper bounds, fixing the Frobenius norm still allows the nuclear norm to increase and move closer to the upper bound, which corresponds to higher intra-sample diversity. There also exist prior works using the nuclear norm to capture two aspects of the output logits signal [1–2].
>
> > The analysis for the connection between the nuclear norm and the optimization utility is also unclear. It is confusing how Eq. (8) provides insights into understanding the correlation between the nuclear/Frobenius norm and the optimization utility. Their correlation is only empirically analyzed in Figure 2. It is unclear whether this correlation holds theoretically.
>
> We find our implementation similar to previous work using nuclear norm for estimation [1]. Actually, it is challenging to mathematically prove that the nuclear norm is strictly linearly correlated with loss reduction or intra-sample diversity due to the complexity and nonlinearity of large models. Instead, we provide some empirical observations as previously mentioned. The exhibited correlations between nuclear norm and both loss reduction & rank suggest using the nuclear norm to characterize intra-sample importance. For loss reduction, our explanation in lines 246–272 tries to provide an intuitive interpretation by decomposing Eq. (8) into two components and analyzing them separately. We have added this theoretical limitation to the manuscript in **Appendix E**.
>
> > It seems that the diversity distance of Eq. (9) focuses only on the distinction from recent data $Q$. It doesn't measure the inter-sample diversity of samples in the current batch $\mathcal{B}_t$. If so, it is possible to choose repeated samples from $\mathcal{B}_t$.
>
> The reviewer's understanding is correct that, in theory, the formulation may select repeated samples within the same batch. However, because the mini-batch size $B=8$ is much smaller than the buffer size $M=1024$ (can be further scaled without notable impact on overall memory/training time consumption; see Appendix B.1) in our experiments, and the data are shuffled, it is unlikely for similar samples to appear within a batch without also appearing in the memory buffer. Thus, this issue is negligible in practice, and the introduction of the large memory buffer is precisely how our method extends beyond prior diversity-estimation approaches, which consider only within-batch diversity and may therefore be biased.

---

> ### Author Response · Authors · 2025-11-27
> **Official Response to Reviewer zKZs from Author (Part II)**
>
> To further validate this, we conducted an additional experiment below (same setting as Table 3 on Qwen-2.5-7B) showing that additionally incorporating intra-batch diversity yields no notable difference.
>
> |Method|MMLU|ScienceQA|GSM8K|HumanEval|
> |:---:|:---:|:---:|:---:|:---:|
> |w/o within-batch diversity|63.34±0.36|95.19±0.22|79.91±0.23|46.28±0.35|
> |w/ within-batch diversity|63.28±0.34|95.26±0.19|79.80±0.38|46.39±0.42|
>
> > Besides, a larger buffer size $M$ might lead to better measurement of the inter-sample diversity, but will also produce high memory requirements. It would be better to show the impact of the value of $M$ in balancing the inter-sample diversity measurement and the memory requirement.
>
> We apologize for not emphasizing this clearly in the maintext. Regarding the requested experiments, the reviewer can refer to **Appendix B.1 (Figures 6 and 7)**. The empirical results show that the memory requirements remain small, and the increase in memory usage with larger $M$ is negligible due to our random down-projection operation (Eq. 10), which is one of our core contributions. We have added a pointer in Section 4.1 directing readers to this analysis in the revised manuscript.
>
> > Some experimental settings are missing, e.g., the value of the buffer size $M$, the trade-off factor $\alpha$, etc.
>
> > (3) What is the value of the trade-off factor in Eq. (11)? How will it affect the online batch selection performance?
>
> > (4) What is the value of $M$ in the experiments?
>
> The buffer size $M$ is set to 1024 across all experiments. The trade-off factor $\alpha$ for Llama-3.1-8B and Qwen-2.5-7B on the four datasets is listed in **Table 6**. We have highlighted this information in **Section 4.1** of the revised manuscript.
>
> For sensitivity analysis of the trade-off factor $\alpha$, the reviewer can refer to **Appendix B.1 (Figure 8)**. A properly chosen $\alpha$ is beneficial for online batch selection performance.
>
> > (1) Why does it require considering the optional history $\mathcal{H}_t$ in the optimization problem of Eq. (1)?
>
> The optional history $\mathcal{H}_t$ corresponds to the previous samples stored in the memory buffer in our method. It is natural to allow decision-making based on previous history in online batch selection, and including it yields a more comprehensive formulation.
>
> > (2) How would the discrete Fourier transform (DFT) matrices $F_1$, $F_2$ be selected?
>
> The DFT matrices $F_1$ and $F_2$ are not explicitly constructed. They are implicitly applied through FFT in our implementation. We use `X = torch.fft.fft(X, dim=-1).real` for the column-side DFT ($F_2$) and `X = torch.fft.fft(X, dim=0).real` for the row-side DFT ($F_1$). Thus, the choice of $F_1$ and $F_2$ simply corresponds to the standard orthonormal DFT basis implemented by the FFT operator.
>
> > (5) How is the representation $z$ obtained in subsection 3.2, given the autoregressive language models in subsection 2.1?
>
> The reviewer can refer to Eq. 10. $z_t^i$ is obtained by bidirectional random down-projection of the logits matrix $L(x_t^i;\theta_t)\in\mathbb{R}^{N\times V}$ generated by the autoregressive language model, followed by a vectorization operation.
>
> **References**
>
> [1] Confidence and Dispersity Speak: Characterizing Prediction Matrix for Unsupervised Accuracy Estimation. ICML2023.
>
> [2] Towards Discriminability and Diversity: Batch Nuclear-norm Maximization under Label Insufficient Situations. CVPR2020.

---

### Official Review · Reviewer_PaeC · 2025-10-31

**Soundness:** 3
**Presentation:** 3
**Contribution:** 2
**Rating:** 4
**Confidence:** 4

**Summary:**

The authors introduce a Utility-Diversity optimized Sampling (UDS) method for SFT, focusing on 3 main axioms for scoring batches live to be chosen for fine-tuning: 1) jointly considering utility, and diversity for both inter-sample and intra-sample inclusion, 2) preventing external access or data leakage, 3) efficient reduction in training time compared to full SFT.

To capture intra-sample value they use the norms of the logits matrix (equation 4),  and for inter-sample diversity the do a bilinear projection of logits for compact embeddings and show matching against historical data reduces redundancy (equation 9). Using the combined score they introduce an algorithm that picks the best batches to be used for SFT. Their technical depth in their proof relies on utilizing a sandwiching argument for relating nuclear norm and Frobenius norm of the logits matrix. They prove a theorem that they can indeed use this sandwiching argument to combine the intra and inter scores they have derived in additive form, while avoiding the storage of an explicit NV × d projection matrix and reduces the computational complexity from O(NV d) to O((N + V ) d log(NV )).

In their experimental setup they use Llama-3.1-8B and Qwen-2.5-7B models as backbone, and test against random sampling, regular (full data), MaxLoss (a scheme prioritizing samples with highest training loss), MaxGrad (a scheme prioritizing samples with largest gradient), RHO-Loss, and GREATS (they claim it as SOTA). Their results show victory over SOTA on all fronts.

One key observable gap in this work is the lack of extensive literature review on this topic. There are a ton of other smart sampling methods for SFT out there, including ones using Fisher Information gain, Optimal Design, and even model based estimations, where other models are used for inferring sample importance. Just one line of work I'm aware of that is heavily relevant is "Data-Efficient Supervised Fine-Tuning of Language Models Using Optimal Design" by Deb et. al. The authors could've done a much more detailed and rigorous literature study on this before claiming win over SOTA. However, the claimed computational gain is not evident from their presented results.

The mathematical rigor of their proof in Appendix C seems reasonable and correct to me, because mainly there is not much complication added other than using the Horn & Johnson Lemma (Lemma 3.1) the right way.

**Strengths:**

The paper is well written, the simplicity they have achieved in conveying their idea is exceptional, and their experimental setup seems to be comprehensive. The work seems original, inspired by their own contribution in ideas. The novelty of combining two aspects of intra and inter diversity is appealing and rare, and significance of their work is in pushing the boundaries of more efficient SFT via introducing more novel ideas.

**Weaknesses:**

Their literature review is not comprehensive and jumps to conclusion on winning over SOTA on all fronts without a deep dive into other methodologies in the literature. It could also benefit from a more comprehensive benchmarking of computation cost gain, since after all this is one of the key aspects where their work, i.e. efficient SFT, is expected to show value. The novelty in their proof technique doesn't seem to be exceptional. It seems to be a modified re-application of Johnson-Lindenstrauss lemma.

**Questions:**

1- Is there a complication in their proof technique that I'm missing? Would love to know more about the challenges they faced in making their proof work.
2- Is there a reason in their limited literature review that I'm not following? Maybe they have clustered the literature review into the best performing ones already, meaning that they already have concluded one of the benchmarks they compare agains is already known to be better than a ton of other works in the literature. In that case it would be great to make that statement clear and elaborate on it a bit more, or otherwise, make sure they include other existing literature as well.

---

> ### Author Response · Authors · 2025-11-27
> **Official Response to Reviewer PaeC from Author (Part I)**
>
> We thank the reviewer for their valuable feedback and provide the following responses.
>
> > One key observable gap in this work is the lack of extensive literature review on this topic. There are a ton of other smart sampling methods for SFT out there, including ones using Fisher Information gain, Optimal Design, and even model based estimations, where other models are used for inferring sample importance. Just one line of work I'm aware of that is heavily relevant is "Data-Efficient Supervised Fine-Tuning of Language Models Using Optimal Design" by Deb et. al. The authors could've done a much more detailed and rigorous literature study on this before claiming win over SOTA.
>
> > Their literature review is not comprehensive and jumps to conclusion on winning over SOTA on all fronts without a deep dive into other methodologies in the literature.
>
> > Is there a reason in their limited literature review that I'm not following? Maybe they have clustered the literature review into the best performing ones already, meaning that they already have concluded one of the benchmarks they compare against is already known to be better than a ton of other works in the literature. In that case it would be great to make that statement clear and elaborate on it a bit more, or otherwise, make sure they include other existing literature as well.
>
> We apologize for the potential misunderstanding regarding the scope of our work. We acknowledge that there are many data selection methods exist for SFT. However, we find that most of them are conducted **offline**, that is, data selection is performed before model training begins. As highlighted at the start of the Introduction, **our work specifically targets the online batch selection setting** (selecting data during training), where existing methods remain very limited.
>
> Additionally, offline and online selection methods serve different purposes and are not directly comparable, particularly in terms of training-time efficiency. Instead, they form a complementary pipeline: offline selection provides a coarse filtering stage to ensure dataset quality, while online selection adaptively captures the changing importance of samples throughout the training process, thereby improving overall efficiency and accuracy [1].
>
> We thank all the reviewers for pointing out several missing references [2–4], and we have added them to the revised manuscript in **Appendix A** under the paragraph "Offline Data Selection for Large Language Models." In **Appendix B.7 (Table 15)**, as suggested by the reviewer, we additionally compare UDS with FisherSFT [4] on the MMLU, ScienceQA, GSM8K, and HumanEval datasets using Qwen-2.5-7B. The results show that UDS also achieves better performance than FisherSFT.
>
> > However, the claimed computational gain is not evident from their presented results.
>
> We suspect the reviewer reached this conclusion by noting that Random Selection and MaxLoss appear more computationally efficient than UDS. However, Random Selection is a trivial baseline that performs no meaningful selection, while MaxLoss does not show notable accuracy improvement over Random Selection. Their accuracy lags far behind UDS. In contrast, when compared with stronger baselines such as RHO-Loss [5] and GREATS [1], both of which achieve significant accuracy gains, UDS demonstrates clear computational advantages, as shown in Table 3.
>
> > It could also benefit from a more comprehensive benchmarking of computation cost gain, since after all this is one of the key aspects where their work, i.e. efficient SFT, is expected to show value.
>
> We agree that more efficiency metrics could further strengthen our claims. For online batch selection, we use "throughput" as metric following GREATS [5]. Another commonly used metric is  "training time". However, training time corresponds one-to-one with the throughput metric, which can be directly computed from throughput and the number of processed samples. Therefore, it does not provide additional information beyond what "throughput" already captures. Could the reviewer point out the specific computational metric that would be beneficial for our manuscript?

---

> ### Author Response · Authors · 2025-11-27
> **Official Response to Reviewer PaeC from Author (Part II)**
>
> > The novelty in their proof technique doesn't seem to be exceptional. It seems to be a modified re-application of Johnson-Lindenstrauss lemma.
>
> > Is there a complication in their proof technique that I'm missing? Would love to know more about the challenges they faced in making their proof work.
>
> We appreciate the reviewer's question. We believe the key contribution of our theoretical analysis lies in the design of decomposing a large projection matrix into two much smaller matrices with **SRFT-style initialization**, which substantially reduces memory requirements while still satisfying the Johnson–Lindenstrauss (JL) lemma.
>
> While bidirectional random projection may appear straightforward when a huge random projection is unacceptable, the commonly used Gaussian initialized random matrices do not satisfy the JL lemma when combined via a Kronecker product. The Kronecker operation breaks the independence structure required for distance preservation. In contrast, we show that SRFT-style initialization does not suffer from this issue and continues to satisfy the JL lemma under the Kronecker product. This derivation corresponds to Eqs. 21–24.
>
> **Reference**
>
> [1] GREATS: Online Selection of High-Quality Data for LLM Training in Every Iteration. NeurIPS2024.
>
> [2] How to train data-efficient llms. Arxiv2024. (AskLLM, Density sampling)
>
> [3] Data-efficient learning via clustering-based sensitivity sampling: Foundation models and beyond. ICML2024. (Clustered Sampling)
>
> [4] FisherSFT: Data-Efficient Supervised Fine-Tuning of Language Models Using Information Gain. ICML2025.
>
> [5] Prioritized Training on Points that are learnable, Worth Learning, and Not Yet Learnt. ICML2022.

---

### Official Review · Reviewer_hMgP · 2025-11-01

**Soundness:** 2
**Presentation:** 2
**Contribution:** 3
**Rating:** 4
**Confidence:** 4

**Summary:**

The paper proposes a online data selection method for fine-tuning LLMs. It operates without a validation set, instead using the model's own signals to identify valuable training instances. The core idea is to select data that promises a large loss reduction (utility) and is different from recently seen samples (diversity). To achieve this, UDS calculates a composite score for each data sample based on two components: intra and inter sample importance scores. The authors evaluated UDS on several benchmarks, including MMLU, ScienceQA, GSM8K, and HumanEval, using Llama-3.1-8B and Qwen-2.5-7B as base models.

**Strengths:**

+ The approach is strongly motivated by the idea that achieving the highest possible model performance is the primary goal.

+ The method is self-contained ("self trace") and does not require a held-out validation set. This makes it more versatile and applicable in scenarios where creating a representative validation set is difficult or impractical.

+ The authors have shown strong performance.

**Weaknesses:**

- The fundamental assumption that a large loss reduction is always beneficial is questionable. Such samples could be outliers, noisy data, or adversarial examples. By prioritizing them, the model might be reinforcing incorrect patterns or learning "bad regions" of the data distribution that are better left under-learned.

- A sample that yields a large loss reduction is not necessarily more informative; it could simply be an "easy" point for the model to learn quickly. The method may not distinguish between truly valuable, complex examples and those that just offer a steep but potentially trivial learning gradient.

- The inter-sample diversity score only compares a candidate sample to a buffer of historical samples. It does not ensure diversity within the newly formed batch. This could lead to selecting a batch of samples that are all novel compared to the past but highly redundant with each other, limiting the learning signal in that update step.

- The method's reliance on in-distribution data for both training and testing means its robustness is unknown. The lack of a validation set is a significant risk when facing domain shift, as the model's internal signals may no longer correlate with generalization performance on unseen distributions.

**Questions:**

Could you elaborate on why reducing repetition within a single training instance is critical for improving model performance?

How does the method distinguish between a valuable, under-learned sample and a noisy or misleading one that also produces a large loss reduction? How do you prevent the model from reinforcing learning in a "bad direction"?

The inter-sample score ensures diversity against past data. What is the motivation for not also considering diversity among the samples selected for the current batch to prevent redundancy in a single gradient step?

Have you evaluated this method when fine-tuning models that have already undergone instruction tuning?

How does your method perform on datasets that require long-context reasoning, where both utility and diversity might be harder to capture?

How does this selection strategy perform when the test set is from a different distribution than the training data?

How much total data was used for the training experiments, and were the different datasets combined or trained on separately?

In Figure 4, you state that K=8 corresponds to full-dataset fine-tuning. Could you explain this, as K usually represents the number of selected samples from a larger batch?

---

> ### Author Response · Authors · 2025-11-27
> **Official Response to Reviewer hMgP from Author (Part I)**
>
> We thank the reviewer for their valuable feedback and we have provided additional experimental results and analysis in the revised manuscript.
>
> > The fundamental assumption that a large loss reduction is always beneficial is questionable. Such samples could be outliers, noisy data, or adversarial examples. By prioritizing them, the model might be reinforcing incorrect patterns or learning "bad regions" of the data distribution that are better left under-learned.
>
> > How does the method distinguish between a valuable, under-learned sample and a noisy or misleading one that also produces a large loss reduction? How do you prevent the model from reinforcing learning in a "bad direction"?
>
> Thank you for your insightful question. We understand that there are many other meaningful data selection criteria (such as filtering out outliers, noisy data, or adversarial examples), but different data selection methods have their own focus according to their specific application scenarios; no single method can address all aspects.
>
> In this work, we focus on online batch selection for SFT, where the candidate data points are usually already carefully curated (SFT datasets are typically small and contain high-quality data), from which "outliers, noisy data, or adversarial examples" have already been filtered. Nevertheless, even with high-quality SFT dataset, the importance of samples can shift throughout the learning process [1]. Compared to offline data selection, the core objective of online batch selection is to **capture these dynamic changes in data importance during training**, which is precisely what our UDS design addresses. If bad samples exist in the original dataset, we can first apply offline techniques to filter out these noisy or outlier points before training begins, since their toxic status usually does not change during learning.
>
> Additionally, we believe that selecting samples with large loss reduction is a widely accepted criterion for capturing sample utility, as many existing data selection methods employ variants of loss reduction [1–3], including those under the online batch selection setting [1].
>
> > A sample that yields a large loss reduction is not necessarily more informative; it could simply be an "easy" point for the model to learn quickly. The method may not distinguish between truly valuable, complex examples and those that just offer a steep but potentially trivial learning gradient.
>
> Thank you for your valuable comment. The most precise interpretation of loss reduction is how much the model can learn **at the current stage**, rather than how much information a sample inherently carries or how complex it is. The online batch selection setting can be viewed as a form of implicit curriculum learning, and many prior work on curriculum learning has shown the benefit of scheduling data in an "easy-to-hard" order [4]. **Thus, if a data point can be learned sufficiently, that is indeed what we expect to select at this stage. For other samples that are more complex, even if they are more informative, if the model is not yet strong enough to effectively capture their patterns (i.e., they are not learnable), they may not be useful for learning at the current stage.** Within the implicit curriculum learning framework, as the model becomes stronger, it gains the ability to capture more complex data points; at that point, these more difficult samples can then be selected for training.
>
> > The inter-sample diversity score only compares a candidate sample to a buffer of historical samples. It does not ensure diversity within the newly formed batch. This could lead to selecting a batch of samples that are all novel compared to the past but highly redundant with each other, limiting the learning signal in that update step.
>
> > The inter-sample score ensures diversity against past data. What is the motivation for not also considering diversity among the samples selected for the current batch to prevent redundancy in a single gradient step?
>
> The reviewer's understanding is correct that, in theory, the formulation may select repeated samples within the same batch. However, because the mini-batch size $B=8$ is much smaller than the buffer size $M=1024$ (can be further scaled without notable impact on overall memory/training time consumption; see Appendix B.1) in our experiments, and the data are shuffled, it is unlikely for similar samples to appear within a batch *without* also appearing in the memory buffer. Thus, this issue is negligible in practice, and the introduction of the large memory buffer is precisely how our method extends beyond prior diversity-estimation approaches, which consider only within-batch diversity and may therefore be biased.

---

> ### Author Response · Authors · 2025-11-27
> **Official Response to Reviewer hMgP from Author (Part II)**
>
> To further support this, we conducted an additional experiment showing that incorporating diversity calculation within the batch yields no notable difference. This simple comparison follows the experimental setting in Table 3 using Qwen-2.5-7B.
>
> |    Method  |    MMLU    | ScienceQA  |   GSM8K    | HumanEval  |
> | :-----: | :--------: | :--------: | :--------: | :--------: |
> | w/o within-batch diversity | 63.34±0.36 | 95.19±0.22 | 79.91±0.23 | 46.28±0.35 |
> | w/ within-batch diversity  | 63.28±0.34 | 95.26±0.19 | 79.80±0.38 | 46.39±0.42 |
>
> > The method's reliance on in-distribution data for both training and testing means its robustness is unknown. The lack of a validation set is a significant risk when facing domain shift, as the model's internal signals may no longer correlate with generalization performance on unseen distributions.
>
> > How does this selection strategy perform when the test set is from a different distribution than the training data?
>
> We appreciate the reviewer's insightful questions. In principle, using a validation set aligned with the OOD test domain can indeed improve robustness under domain shift. **However, relying on a validation set can be a double-edged sword that introduces several practical limitations:**
>
> 1. First, in realistic training settings, one usually does not know the exact distribution from which the downstream test data will be drawn, so the applicability of validation-set-based methods may be limited.
>
> 2. Second, the performance of validation-set-based methods may heavily rely on the size of the validation set. If the validation set is small, it may not sufficiently capture the downstream distribution; there may be large bias in evaluation simply because the data sample differs from the validation set. If the validation set is large, it may become computationally expensive if we evaluate the model on it frequently (e.g. per batch).
>
> To further evaluate the robustness of UDS, we additionally include an experiment where we train on GSM8K and test on OOD datasets such as MATH500 and AMC23. Encouragingly, UDS continues to exhibit strong performance across both in-distribution and OOD settings, demonstrating robustness to distribution shifts. For detailed results, we refer the reviewer to **Appendix B.5 (Table 14)**.
>
> > Could you elaborate on why reducing repetition within a single training instance is critical for improving model performance?
>
> Thanks for the valuable question. We view reducing repetition within a single training instance as a way to enhance "token-level diversity". Since the goal of SFT is essentially to "memorize" token sequences, insufficient token-level diversity may cause the model to overfit to a small set of frequently repeated tokens, thereby limiting the diversity of its output patterns.
>
> For example, in a bad case where a sample contains many repeated copies of the same sentence (i.e., very low token-level diversity), the model may learn to produce repetitive outputs as well, which is clearly harmful. The importance of maintaining token-level diversity is conceptually similar to the motivation behind text data augmentation [5] in literature.
>
> > Have you evaluated this method when fine-tuning models that have already undergone instruction tuning?
>
> Thanks for your suggestion. We additionally include an experiment on MMLU, ScienceQA, GSM8K, and HumanEval using **Qwen-2.5-7B-Instruct** in **Appendix B.3 (Table 12)**. We observe that UDS continues to show strong effectiveness in this setting.
>
> > How does your method perform on datasets that require long-context reasoning, where both utility and diversity might be harder to capture?
>
> Thanks for your suggestion. We further add an experiment using **Qwen-2.5-7B on the MATH dataset** in **Appendix B.4 (Table 13)**, where the maximum sequence length is set to **2048** (compared to 512 in previous experiments). The results demonstrate that UDS remains more effective than other baselines in this challenging long-context scenario, confirming its ability to capture both utility and diversity even in long-context reasoning tasks.
>
> > How much total data was used for the training experiments, and were the different datasets combined or trained on separately?
>
> We are happy to clarify these points for the reviewer.
>
> 1. For the first question, **Table 15** in **Appendix D.1** reports the total number of training samples for each dataset. The fraction of online-selected data used in each experiment is shown in **Table 5**.
>
> 2. For the second question, each dataset was trained **separately**. Specifically:
>
>     – MMLU results were trained on MMLU’s auxiliary dataset and tested on its test set;
>
>     – GSM8K results were trained on GSM8K’s training set and tested on its test set;
>
>     – ScienceQA results were trained on its training set and tested on its test set;
>
>     – HumanEval results were trained on CodeAlpaca and evaluated on the HumanEval benchmark.

---

> ### Author Response · Authors · 2025-11-27
> **Official Response to Reviewer hMgP from Author (Part III)**
>
> > In Figure 4, you state that K=8 corresponds to full-dataset fine-tuning. Could you explain this, as K usually represents the number of selected samples from a larger batch?
>
> We are pleased to clarify this for the reviewer. In our experiments, the mini-batch size is fixed at **B = 8** throughout training. Under the online batch selection setting, we select **K out of B** samples in each mini-batch for training. In Figure 4 of the original submission (Figure 5 in the revised version), we examine how different selection ratios affect accuracy by varying **K from 1 to 8**. Therefore, when **K = 8**, we select all samples in the mini-batch that actually perform no data selection. Hence, we state that **K = 8 corresponds to full-dataset fine-tuning within each mini-batch**.
>
> **Reference**
>
> [1] GREATS: Online Selection of High-Quality Data for LLM Training in Every Iteration. NeurIPS2024.
>
> [2] LESS: Selecting Influential Data for Targeted Instruction Tuning. ICML2024.
>
> [3] Estimating Training Data Influence by Tracing Gradient Descent. NeurIPS2020.
>
> [4] A Survey on Curriculum Learning. TPAMI2021.
>
> [5] Unsupervised Data Augmentation for Consistency Training. NeurIPS2020.

---

### Official Review · Reviewer_Ey6P · 2025-11-01

**Soundness:** 3
**Presentation:** 4
**Contribution:** 3
**Rating:** 6
**Confidence:** 3

**Summary:**

The paper proposes **UDS (Utility-Diversity Sampling)**, an online batch selection framework for SFT of LLMs. For each candidate example, UDS computes (i) an **intra-sample score**: the **nuclear norm** of the sequence-by-vocab logits matrix to capture both optimization utility and within-sequence diversity; and (ii) an **inter-sample score**: a **diversity distance** computed in a low-dimensional embedding against a FIFO memory of recently selected samples. The two scores are combined to select the top-K examples per batch without extra backprop or external resources. Experiments on MMLU, ScienceQA, GSM8K, and HumanEval with Llama-3.1-8B and Qwen-2.5-7B report consistent gains and, in some settings, higher throughput than full-dataset SFT.

**Strengths:**

- **Clear, simple selection signal**: Using the logits nuclear norm avoids gradient computation and external models/val sets, aligning with the forward pass that already occurs each step. The paper also motivates the nuclear norm via bounds with the Frobenius norm.
- UDS consistently outperforms baselines (MaxLoss/MaxGrad/RHO-Loss/GREATS) on accuracy and often improves throughput vs. full-data SFT.

**Weaknesses:**

- All fine-tuning uses LoRA with small batch sizes; it would help to see results under full-parameter SFT or larger batches to ensure the gains persist when per-step signal-to-noise changes.
- ablations on buffer update policies (e.g., reservoir sampling, class-aware sampling) are not shown.
- Some missing recent papers on data efficiency for LLMs for related works discussion:

    1. Sachdeva, N., Coleman, B., Kang, W.-C., Ni, J., Hong, L., Chi, E. H., Caverlee, J., and Cheng, D. Z. How to train data-efficient llms. (AskLLM, Density sampling)

    2. Axiotis, K., Cohen-Addad, V., Henzinger, M., Jerome, S., Mirrokni, V., Saulpic, D., Woodruff, D. P., and Wunder, M. Data-efficient learning via clustering-based sensitivity sampling: Foundation models and beyond. In Proceedings of the 41st International Conference on Machine Learning. PMLR, 2024. (Clustered Sampling)

    3. Deb et al., FisherSFT: Data-Efficient Supervised Fine-Tuning of Language Models Using Information Gain. In Proceedings of the Forty-Second International Conference on Machine Learning (ICML), 2025.

**Questions:**

- Have you tried alternative **buffer policies** (reservoir sampling; clustering-based prototypes) and distance aggregations (e.g., farthest-k, soft-min) to reduce sensitivity to buffer composition?
- The first-order analysis is written for SGD. Is the loss-reduction correlation under Adam with LoRA adapters?
- The SRFT-style factorization - did you compare to smaller random features (e.g., sparse JL, CountSketch-style tensor sketches)

---

> ### Author Response · Authors · 2025-11-27
> **Official Response to Reviewer Ey6P from Author**
>
> We thank the reviewer for the valuable feedback and believe the detailed responses below sufficiently address the concerns.
>
> > All fine-tuning uses LoRA with small batch sizes; it would help to see results under full-parameter SFT or larger batches to ensure the gains persist when per-step signal-to-noise changes.
>
> Thank you for the suggestion. We have conducted additional two experiments using full-parameter SFT as well as larger batch sizes, respectively. The results are provided in **Appendix B.3 (Tables 10 and 11)**. Across all settings, UDS consistently outperforms the corresponding baselines, further demonstrating the robustness and generalization capability of our method.
>
> > ablations on buffer update policies (e.g., reservoir sampling, class-aware sampling) are not shown.
>
> > Have you tried alternative buffer policies (reservoir sampling; clustering-based prototypes) and distance aggregations (e.g., farthest-k, soft-min) to reduce sensitivity to buffer composition?
>
> We appreciate the reviewer's suggestion. We have incorporated additional ablation studies on both buffer update policies and distance aggregation strategies in **Appendix B.2 (Tables 7 and 8)**. The results indicate that UDS is not sensitive to these design choices. Our default configuration, using a FIFO buffer and average-distance aggregation, already suffices to capture inter-sample diversity without requiring more sophisticated mechanisms.
>
> > Some missing recent papers on data efficiency for LLMs for related works discussion:
>
> Thank you for pointing out these relevant works [1-3]. These papers focus on offline data selection for LLMs, and we have added them to the revised manuscript in **Appendix A** under the paragraph "Offline Data Selection for Large Language Models."
>
> In **Appendix B.7 (Table 15)**, we additionally compare UDS with FisherSFT [3] on the MMLU, ScienceQA, GSM8K, and HumanEval datasets using Qwen-2.5-7B. The results show that UDS also achieves better performance than FisherSFT.
>
> > The first-order analysis is written for SGD. Is the loss-reduction correlation under Adam with LoRA adapters?
>
> We appreciate the reviewer's question. The empirical results in Figure 2 regarding the loss-reduction correlation were obtained using Adam, which is also the optimizer used in all our experiments. In addition, we have derived the first-order analysis for Adam in **Appendix C.2**, and the resulting formulation differs only slightly from the SGD version. The overall implications of the analysis remain unchanged.
>
> > The SRFT-style factorization - did you compare to smaller random features (e.g., sparse JL, CountSketch-style tensor sketches)
>
> Thank you for the suggestion. We have added these comparisons in **Table 9**. We find that SRFT-style and CountSketch-style tensor sketches perform similarly well, while Sparse JL performs significantly worse. This can be attributed to the fact that Sparse JL fails to satisfy the JL lemma in our bidirectional random matrix construction, similar to the Gaussian construction, because the Kronecker product disrupts the independence structure of the random variables.
>
> Since the bilinear down-projection with SRFT-style factorization already introduces negligible computational & memory overhead, we believe it is sufficient for practical use. A more detailed analysis is provided in **Appendix B.2**.
>
> **Reference**
>
> [1] How to train data-efficient llms. Arxiv2024. (AskLLM, Density sampling)
>
> [2] Data-efficient learning via clustering-based sensitivity sampling: Foundation models and beyond. ICML2024. (Clustered Sampling)
>
> [3] FisherSFT: Data-Efficient Supervised Fine-Tuning of Language Models Using Information Gain. ICML2025.

---

### Author Response · Authors · 2025-11-27
**Sincerely Looking Forward to the Reviewers' Feedback**

Dear Reviewers,

We sincerely appreciate the depth and thoughtfulness of your comments, and we greatly value the time and effort you have dedicated to evaluating our work.

We have provided detailed responses to all reviewer questions and have updated the manuscript accordingly, with **all revisions highlighted in blue** throughout the paper.

If you have any further questions or require additional clarification, please feel free to let us know. We are grateful for the opportunity to continue the discussion if needed.

Best regards,

The Authors

---

### Meta-Review · Area_Chair_4KAr · 2026-01-03

**Summary:**

This paper studies online batch selection for supervised fine-tuning (SFT). The key idea is to score examples in each incoming batch and then use the top-$K$ examples for fine-tuning. The proposed algorithm is a heuristic, which is partially analyzed. The algorithm is evaluated on four tasks and compared to six baselines. The paper is well written and empirical results are good. The reviewers had many concerns, which the authors addressed in the rebuttal by extensive additional experiments. My main concerns are the novelty and relation to prior works. In short, the proposed approach is a heuristic, which is closely related to prior works that solve the problem arguably better. The paper requires a major revision to take this into account. In particular:

* **Online batch selection:** The authors argue that this problem is novel and I agree. However, the main algorithmic contribution (lines 4-10 in Algorithm 1) is a standard active learning problem where each batch is treated as an offline dataset.

* **Optimal designs:** Since the inner loop of Algorithm 1 is a classic active learning problem, techniques like [optimal experimental designs](https://en.wikipedia.org/wiki/Optimal_experimental_design) can be used to choose most informative examples. The authors should look at [FisherSFT: Data-Efficient Supervised Fine-Tuning of Language Models Using Information Gain](https://proceedings.mlr.press/v267/deb25a.html), which two reviewers point to, and references in this paper for details. Their algorithm FisherSFT can be used to replace lines 4-10 in Algorithm 1 and would be analyzable. The algorithm would be related because it works on token history embeddings, which induce the logits used in this paper. In summary, this paper does an optimal experimental design / Fisher information matrix optimization without being aware of 50+ years of work on this topics, which originate in classic statistics.

This paper requires a major revision to be put in the context of prior works, which are expected to solve online batch selection comparably to the proposed method and with better theoretical guarantees.

**Reviewer Concerns:**

The rebuttal was extensive and included many experiments. However, the work was not put properly in the context of prior works. See my summary for details.

**Reviewer Scores:**

This is hard to estimate since there was no reviewer engagement. The rebuttal was extensive but it did not address the main issue. The main issue is that the main algorithmic contribution is a classic active learning problem that can be arguably solved better.

---

### Decision · Program_Chairs · 2026-01-26

Reject